# PartNeXt: A Next-Generation Dataset for Fine-Grained and Hierarchical 3D Part Understanding

**Penghao Wang   Yiyang He   Xin Lv   Yukai Zhou**
**Lan Xu   Jingyi Yu   Jiayuan Gu**[*]
ShanghaiTech University
https://authoritywang.github.io/partnext

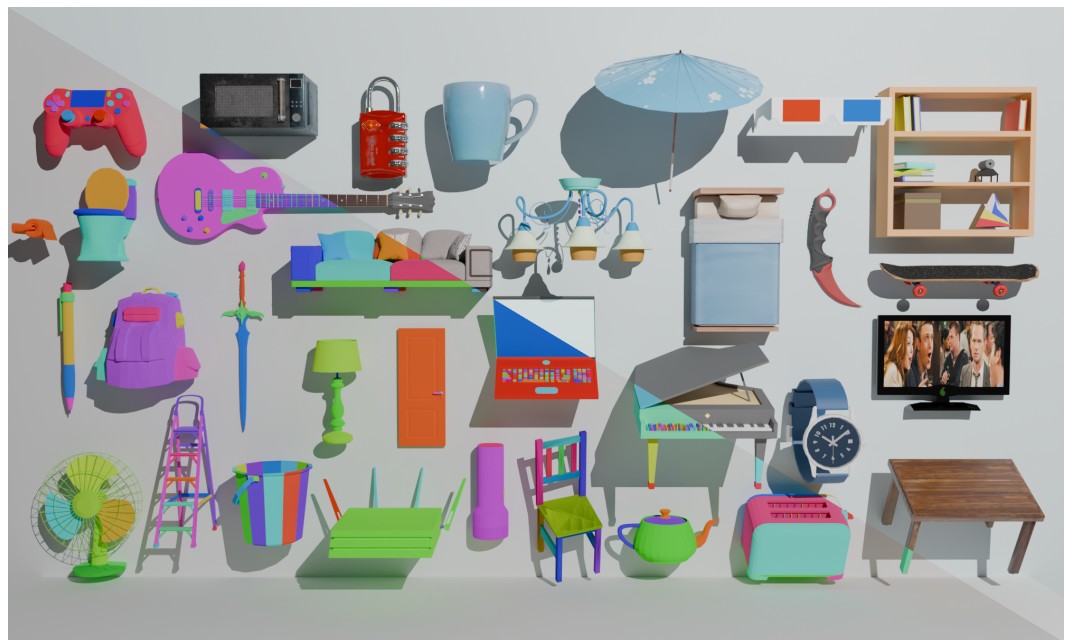

Figure 1: We present **PartNeXt**, a next-generation dataset tailored for fine-grained, hierarchically structured 3D part understanding.

## Abstract

Understanding objects at the level of their constituent parts is fundamental to advancing computer vision, graphics, and robotics. While datasets like PartNet have driven progress in 3D part understanding, their reliance on untextured geometries and expert-dependent annotation limits scalability and usability. We introduce PartNeXt, a next-generation dataset addressing these gaps with over 23,000 high-quality, textured 3D models annotated with fine-grained, hierarchical part labels across 50 categories. We benchmark PartNeXt on two tasks: (1) class-agnostic part segmentation, where state-of-the-art methods (e.g., PartField, SAMPart3D) struggle with fine-grained and leaf-level parts, and (2) 3D part-centric question answering, a new benchmark for 3D-LLMs that reveals significant gaps in open-vocabulary part grounding. Additionally, training Point-SAM on PartNeXt yields substantial gains over PartNet, underscoring the dataset's superior quality and diversity. By

[*]Corresponding Author.

39th Conference on Neural Information Processing Systems (NeurIPS 2025) Track on Datasets and Benchmarks.

combining scalable annotation, texture-aware labels, and multi-task evaluation, PartNeXt opens new avenues for research in structured 3D understanding.

# 1 Introduction

Understanding objects at the level of their constituent parts is fundamental to numerous tasks in computer vision, computer graphics, and robotics. From semantic and instance-level segmentation [26, 53, 48, 25] to generative modeling [18, 22] and robot manipulation [45, 42], fine-grained part reasoning enables models to infer structure, affordance, and function in ways that align closely with human perception and action. As David Marr famously argued in his theory of vision [30], the construction of intermediate representational primitives—such as parts—is a critical step in transforming raw sensory data into higher-level perceptual and behavioral intelligence.

The PartNet dataset [32] has driven progress in part-level 3D understanding. It provides over 573K part annotations across 26K 3D models spanning 24 object categories, organized in expert-defined hierarchies. It has catalyzed a wave of research in 3D part segmentation [27, 54], affordance analysis [6, 50], and retrieval-based generation [17, 21]. Despite its impact, several limitations in data quality and the annotation tool hinder its broader usability and scalability. **First, some annotations in PartNet require remeshing the mesh, which may lead to missing textures on some objects and deformation of the mesh geometry.** This restricts the availability of visual cues such as color and material, which are often essential for both human annotators and learning-based models to accurately recognize and segment object parts. As a result, many existing methods [39, 26] either rely solely on geometry or operate on small textured subsets like PartNet-Mobility [44]. **Second, its annotation interface demands significant expertise in 3D modeling, presenting a barrier to crowdsourcing and large-scale expansion.** For instance, annotators have to manually draw curves to cut meshes and carefully inspect sliced cross-sections to identify interior components—procedures that are time-consuming and unintuitive for non-experts.

Creating a new 3D part-level annotated dataset is both necessary and highly challenging. While there has been notable progress in annotating 2D multi-granular masks [15], part-level annotation in 3D introduces a unique set of difficulties. Unlike 2D images, 3D objects can contain complex interior structures, making segmentation significantly more demanding. First, designing an intuitive and easy-to-use annotation interface for non-expert users is non-trivial—especially when annotating interior or occluded parts. Second, incentivizing annotators to produce useful and fine-grained segmentations requires thoughtful task design and guidance, as well as mechanisms to ensure consistency and quality. Third, if hierarchical labels are desired, defining a coherent and extensible taxonomy that generalizes across categories remains an open problem.

In this work, we introduce **PartNeXt**, a next-generation dataset for fine-grained and hierarchical 3D part understanding. PartNeXt contains 23,519 high-quality, textured meshes annotated with detailed part masks spanning 50 object categories. The models are sourced from Objaverse [5], ABO [3], and 3D-FUTURE [8], ensuring a wide diversity of appearances and geometries while covering several widely used 3D object datasets. **To support efficient and scalable annotation, we develop a fully web-based interface tailored for crowdsourcing.** The interface features a dual-panel layout: one panel displays the unannotated regions of the mesh, while the other shows parts that have already been annotated. Annotators assign parts by selecting faces from the unannotated panel and moving them into the annotated panel. This intuitive and visually guided workflow minimizes the need for specialized 3D expertise and significantly boosts annotation throughput, especially for complex objects with interior structures. Importantly, PartNeXt annotations are performed directly on textured meshes. To enable this, we implement custom algorithms for visualizing partially segmented, high-resolution textured meshes and provide a multi-granular suite of face-based selection tools that enhances annotation flexibility, efficiency, and precision. **We further enhance scalability and consistency by integrating AI tools into the annotation workflow.** Specifically, we use CLIP-based[37] filtering to select high-quality, category-consistent assets and employ GPT-4o[13] to bootstrap part hierarchies across categories. The resulting dataset supports both high annotation quality and broad taxonomic coverage.

To showcase the utility of our dataset, we first evaluate performance on class-agnostic part segmentation, which assesses a model's ability to identify and segment semantically meaningful parts without relying on category-specific priors. We find that state-of-the-art methods such as PartField [25], SAM-

Part3D [48], and SAMesh [39] perform noticeably worse on our fine-grained dataset, particularly when segmenting leaf-level parts in the hierarchy. In addition, we introduce a new benchmark: 3D part-centric question answering. This task is tailored for 3D Large Language Models (3D LLMs), and evaluates their ability to perform open-vocabulary part grounding, detection, and question answering. We benchmark leading 3D LLMs, including ShapeLLM [35], 3D-LLM [11], and PointLLM [46], and observe that current models struggle with part-centric queries—underscoring the complexity and significance of this challenge. Lastly, we train the interactive segmentation model Point-SAM [53] on PartNeXt, which substantially outperforms its counterpart trained on PartNet. It demonstrates the enhanced quality, diversity, and utility of our dataset for fine-grained 3D understanding.

## 2 Related Work

**3D Datasets**   Large-scale 3D repositories have played a pivotal role in advancing graphics, vision, and robotics. ShapeNet [1] is one of the earliest efforts, aggregating a large collection of textured meshes organized under WordNet synsets. Its core subset provides about 51K models with filtered mesh and texture quality, and has been widely used for 3D shape analysis [33, 34, 55] and synthetic data generation [31, 7, 9]. More recently, Objaverse [5] expanded the scale of 3D data to over 800K richly tagged objects sourced from Sketchfab. Objaverse-XL [4] further expands this effort.

In addition to broad-category collections, several datasets target specific domains. Amazon-Berkeley Objects (ABO) [3] includes 7,953 artist-designed textured meshes accompanied by product images and metadata from Amazon. 3D-FUTURE [8] offers approximately 16K industrial CAD furniture models with high-resolution textures created by professional designers, accompanied by photorealistic synthetic renderings. Thingi10K [52] curates 10K 3D-printable models from Thingiverse. Apart from mesh-based datasets, multi-view image collections have been used to reconstruct 3D shapes from real-world observations, like CO3D [38], MVImgNet [51], OmniObject3D [43].

**3D Part Annotation**   Despite many existing 3D object datasets, part-level annotations remain limited due to high annotation costs, the inherent complexity of 3D labeling, and underexplored interface design. PartNet [32] pioneered large-scale fine-grained, hierarchical part annotations, providing detailed part masks for approximately 26K shapes across 24 categories. It enabled benchmarks in semantic, hierarchical, and instance-level part segmentation. Prior to PartNet, ShapeNet-Part[49] offered coarse part labels for 16 categories from ShapeNet, and remains a popular benchmark for semantic part segmentation. Other efforts include Fusion360 [41], which focuses on CAD models and B-Rep. Several extensions of PartNet have further enriched the landscape. PartNet-Mobility[44] adds kinematic joint annotations to over 2K articulated objects, supporting research on articulated object understanding [18, 22, 17, 21]. GAPartNet[10] redefines part semantics by grouping them according to functionality. 3D AffordanceNet [6] augments selected PartNet shapes with point-wise probabilistic affordance scores to support functional reasoning. In addition to synthetic datasets, part-level annotations have also been introduced in real-world scanned data, such as ScanObjectNN [40] and AKB-48 [24]. Recently, PartObjaverse-Tiny [48] was introduced to evaluate open-vocabulary part segmentation, comprising 200 complex 3D objects with semantic and instance annotations.

Our new dataset, PartNeXt, offers part annotations across a broader range of categories than PartNet (50 vs 24). We introduce a novel, web-based annotation interface designed for efficient labeling, particularly of interior structures. Unlike PartNet, which provides separate, untextured part meshes, PartNeXt directly annotates parts on textured meshes. This avoids common issues such as the need for extra alignment between ShapeNet and PartNet when textures are required.

**3D Part Understanding**   Part annotations enable many downstream tasks relevant to 3D part understanding, such as part segmentation, part assembly [20, 12, 19], part-based generative models [18, 22], and articulated object reconstruction [14, 23, 29, 2]. PartNet has proposed a detection-by-segmentation method to address instance part segmentation and benchmarked a set of close-vocabulary segmentation methods on semantic part segmentation. Recently, there has been growing interest in open-vocabulary part segmentation. PartField [25] learns feature embeddings and applies clustering to generate segmentations. SAMesh [39] combines multi-view SAM guidance with a tailored connectivity detection algorithm to produce high-quality segmentations. Point-SAM [53], following SAM [15], extends interactive segmentation to 3D point clouds.

Besides, recent efforts have extended large-scale vision-language models to support spatial reasoning in 3D environments. Models such as PointLLM [46], ShapeLLM [35], 3D-LLM [11], and GPT4Point [36] aim to align 3D point clouds with language queries, enabling tasks such as object classification, 3D captioning, and functionality understanding.

# 3 Data Annotation

In this section, we present the data annotation process of **PartNeXt**, a next-generation 3D dataset with fine-grained and hierarchical part annotations. We detail our data collection and preprocessing pipeline (Sec. 3.1), the construction of consistent and functionality-aware part hierarchies (Sec. 3.2), and the design of our scalable annotation system (Sec. 3.3) optimized for both efficiency and accuracy. Sec. 3.4 describes comprehensive statistics that highlight the scale, diversity, and richness of PartNeXt.

## 3.1 Data Collection and Preprocessing

We collected high-quality 3D models from several large-scale public datasets, including Objaverse [5], ABO [3], and 3D-FUTURE [8]. Among these, ABO and 3D-FUTURE primarily focus on household furniture CAD models and provide reliable category annotations, allowing straightforward filtering to select relevant categories. In contrast, Objaverse spans a much broader range of categories and exhibits greater variation in quality, including a large number of 3D scans and multi-object scenes, which makes consistent category annotation significantly more challenging.

To curate a clean subset from Objaverse, we first applied a series of filters based on metadata. Specifically, we removed: (1) models containing animation, (2) models with more than 130k faces, and (3) models tagged as scans or architectural objects. Since Objaverse does not provide explicit category annotations, we adopted a text similarity-based classification approach to label and further filter the remaining high-quality models.

We began by defining a set of around 100 common object categories. Using CLIP's text encoder, we encoded both the category names and the descriptive captions provided by Cap3D [28] for each object. Each object was assigned the category whose embedding had the highest cosine similarity to its caption. To ensure label reliability, we discard any object whose highest similarity score was below 0.75. Finally, we selected the 50 categories with the largest number of objects.

## 3.2 Hierarchy Definition and Example Generation

Parts are typically organized in hierarchical structures, reflecting different levels of granularity based on functionality or semantics. A well-defined hierarchy not only facilitates the interpretation of part semantics and functionality, but also improves annotation accuracy by providing annotators with clear and consistent guidance. Following the strategy introduced in PartNet [32], we predefined a structured part hierarchy for each category, along with detailed definitions and illustrative examples for all part nodes within the hierarchy.

In PartNet, part hierarchies were defined by experts based on several criteria, including being *well-defined, consistent, compact, hierarchical, atomic, and complete*. However, these principles were implied implicitly within expert-designed templates. Notably, part hierarchies can also be informed by an object's manufacturing process and intended functionality. To better capture these aspects, we refine and formalize the hierarchy design criteria as follows:

- **Functionality-aware**: Top-level components should consist of the largest indivisible, functionally meaningful parts.

- **Hierarchical**: Deeper levels should be composed of sub-parts as defined during the manufacturing process.

- **Exhaustive variants**: When a part has multiple variants, all possible types should be enumerated under the same parent node to explicitly differentiate between them.

- **Atomicity**: Leaf nodes should represent all parts that cannot be further subdivided.

- **Consistency**: Parts with the same function and structure should be defined consistently across different object categories.

Manually defining detailed hierarchical structures—particularly when enumerating diverse part variants—is a challenging and labor-intensive task. To address this, we leverage GPT-4o [13] to assist in generating hierarchies for each category. Our prompts incorporate the aforementioned design principles to guide the model in producing coarse hierarchies. To further improve coverage of diverse part variants (e.g., a chair may have a foot base or a pedestal base), we collect rendered images for each category and provide GPT-4o with them to refine hierarchies. All AI-generated hierarchies are subsequently reviewed and refined by human experts to ensure accuracy and consistency.

Fine-grained part annotation often involves highly specialized terminology, which can lead to ambiguity during the labeling process. To address this, we provide visual examples for each part to aid annotators. Leveraging the image generation capabilities of GPT-4o, we produce labeled reference images for each part node. These generated samples are manually reviewed, and when necessary, inaccurate or missing examples are supplemented or replaced with high-quality images curated from online sources to ensure clear and faithful visual representations of each component.

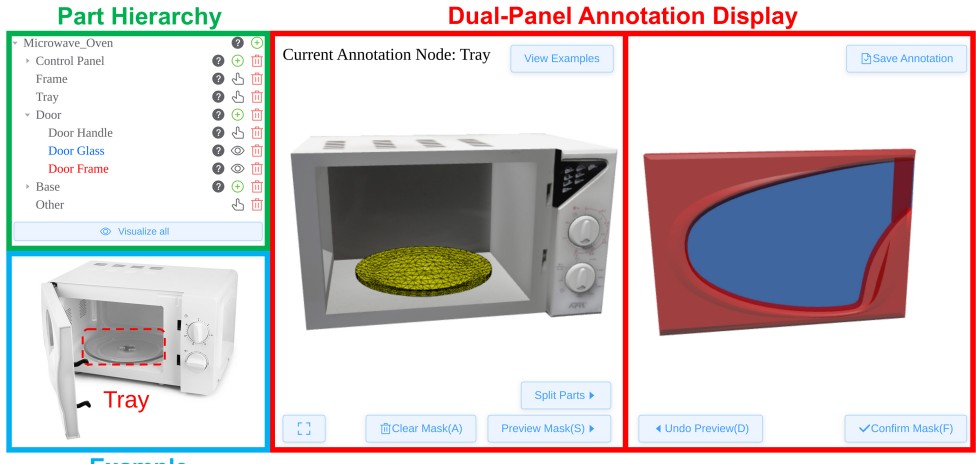

Figure 2: **Illustration of our annotation interface**. The example shows a microwave containing an internal tray. The dual-panel layout allows annotators to first label external parts such as the "door" (as shown in the right panel with already segmented meshes), and then proceed to annotate internal components like the "tray" (visible in the unsegmented mesh in the left panel). This design effectively mitigates occlusion issues during annotation.

## 3.3 Annotation System Design

Our annotation system is fully web-based to support scalable crowdsourcing (see Figure 2). It is designed with three key features to enhance annotation efficiency and quality: (1) a hierarchical annotation workflow, (2) a dual-panel interface, and (3) a suite of selection tools. Please refer to the supplementary video for a demonstration of our annotation interface.

**Hierarchical annotation workflow** We adopt a hierarchical annotation workflow to guide annotators in labeling complex 3D structures. The panel presents a collapsible tree structure representing the predefined part hierarchy. By default, only top-level components are visible; annotators progressively expand the tree and label leaf nodes. This progressive design helps disambiguate multiple part instances with the same semantics but different parent nodes—for example, distinguishing between a cushion on the backrest and one on the seat of a chair. For uncommon or unexpected components not captured by the predefined hierarchy, annotators can create an "Other" node at any level.

**Dual-panel interface** The main interface features a dual-panel layout: the left panel displays the unsegmented 3D mesh, while the right panel shows the segmented result from the same viewpoint. Annotators interactively select mesh faces corresponding to the currently active part node, and once confirmed, the selected region is transferred from the left panel to the right. Each annotated part is assigned a unique color, synchronized with its corresponding node in the hierarchy tree. The dual-panel design is particularly useful for visualizing and labeling interior parts that may be initially occluded in the unsegmented view.

**Selection tools** Previously, PartNet supported selecting mesh subgroups or provided mesh-cutting tools to split parts on a remeshed surface. However, manual mesh cutting is often time-consuming and require experience with mesh processing. In contrast, we ask annotators to work directly on the original textured mesh, and provide three face-selection tools tailored for different use cases:

- **Connected Component Selection:** We provide a connectivity-based selection tool that enables annotators to select mesh regions by simply clicking on a face. The tool automatically selects the entire connected component containing the clicked face. To accommodate meshes with duplicated vertices, we offer two modes for computing connectivity—one that considers exact mesh topology and another that merges nearby vertices—ensuring robust performance across diverse mesh qualities.

- **Bounding-Box selection:** We include a bounding-box selection tool that selects all mesh faces within a user-defined 2D bounding box projected from the current viewpoint. This tool is particularly effective for quickly selecting coherent regions on over-segmented or highly detailed meshes.

- **Per-Face Selection:** For precise control, manual face-by-face selection is supported, allowing annotators to directly add or remove individual faces, offering fine-grained control over the selected regions and complementing the coarser selection tools.

Compared to the selection tools provided in PartNet, our system supports flexible combinations of different selection modes. For example, annotators can first select a connected component and then refine the selection by removing a subset of faces. By operating directly at the face level on textured meshes, our approach eliminates the need for remeshing and preserves the original texture information.

## 3.4 Statistic

We hired 35 professional annotators, and 5 top-performing annotators responsible for data verification and quality control. The annotation backend was deployed on a central server equipped with dual Intel Xeon 6326 CPUs, while annotators operated on consumer-grade PCs powered by Intel Core i5-12400F processors. Prior to labeling, all annotators completed a two-day training session using a curated set of test models to ensure consistency and accuracy. On average, each 3D model required approximately 5 to 6 minutes to annotate.

Our constructed dataset, PartNeXt, provides 350187 annotated instances for 23,519 objects across 50 categories. Specifically, 14,811 instances were sourced from Objaverse, 2,633 from ABO, and 6,075 from 3DFuture. Detailed statistics of the dataset are presented in Table 1. For more statistics, please refer to the appendix. Each annotation underwent at least one review, with a total of 5,211 corrections made. The maximum number of corrections for a single annotation reached 8. The defined part hierarchy spans from a minimum depth of 4 to a maximum depth of 10.

| | All | Axe | Bag | Bed | Bookcase | Bottle | Buck | Cam | Chair | Chandeller | Mouse | Control | Cup | Door | Fan | Flashlight | F-Lamp |
|---|---|---|---|---|---|---|---|---|---|---|---|---|---|---|---|---|---|
| #S | 23519 | 1628 | 69 | 1454 | 574 | 1204 | 168 | 8 | 4277 | 1355 | 129 | 78 | 361 | 9 | 93 | 118 | 78 |
| #P | 350187 | 7142 | 589 | 33661 | 15018 | 4196 | 1974 | 81 | 60819 | 51678 | 520 | 1640 | 1056 | 48 | 950 | 715 | 718 |

| | Fork | Glass | Guitar | Ham. | H-phone | Keyboard | Knife | Lamp | Lap. | Micro. | Monitor | Mug | P-lock | Pen | Piano | Pickaxe |
|---|---|---|---|---|---|---|---|---|---|---|---|---|---|---|---|---|
| #S | 76 | 291 | 279 | 583 | 136 | 228 | 998 | 950 | 94 | 67 | 332 | 498 | 18 | 146 | 106 | 124 |
| #P | 487 | 2833 | 9574 | 2843 | 1225 | 22045 | 3815 | 11073 | 9385 | 1042 | 1941 | 1133 | 83 | 864 | 6945 | 560 |

| | Plier | Sciss. | Screw | Shovel | Skate. | Sofa | Spoon | S-Ladder | Sword | Table | Teapot | Toast. | Toilet | Umb. | Wash. | Watch | WiFi | Wrench |
|---|---|---|---|---|---|---|---|---|---|---|---|---|---|---|---|---|---|---|
| #S | 65 | 40 | 71 | 73 | 141 | 3139 | 101 | 22 | 151 | 2326 | 348 | 28 | 137 | 65 | 19 | 115 | 8 | 141 |
| #P | 523 | 229 | 218 | 525 | 2461 | 48135 | 285 | 433 | 647 | 31836 | 3180 | 549 | 1133 | 1111 | 280 | 1256 | 123 | 610 |

Table 1: **PartNeXt Dataset Statistic**. #S represent number of annotated objects, while #P as the number of total annotated parts.

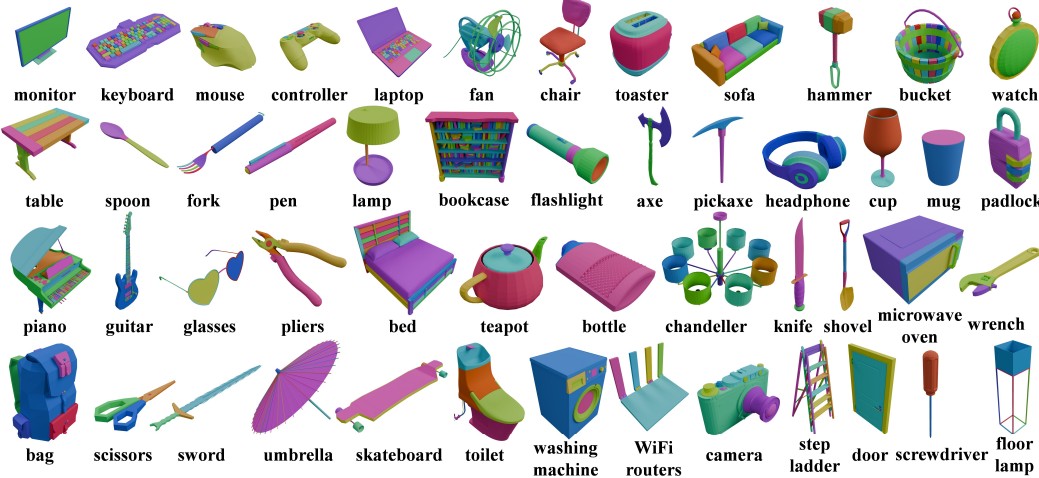

Figure 3: **PartNeXt dataset**. We visualize example shapes with fine-grained part annotations for the 50 object categories in PartNeXt.

# 4 BenchMark and Experiments

Based on our PartNeXt dataset, we introduce two benchmarks: **class-agnostic 3D part instance segmentation** and **part-centric 3D question answering**. Additionally, we examine the quality of our dataset by training a transformer-based interactive 3D segmentation method, and show superior results compared to PartNet.

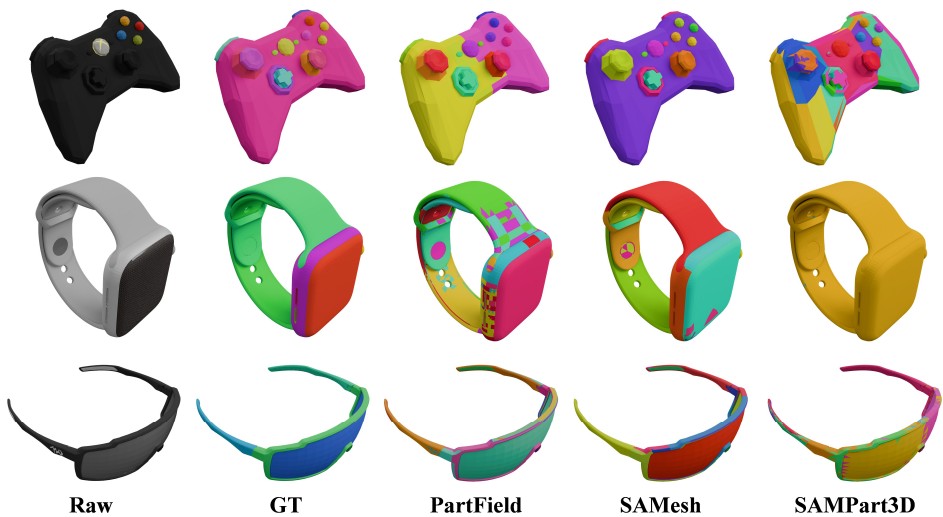

Figure 4: **Part Segmentation Results on PartNeXt**. PartField struggles to separate connected regions, SAMesh excels at fine-grained segmentation but over-segments, while SAMPart3D lacks continuity in weak textures and granularity control.

## 4.1 Class-Agnostic 3D Part Instance Segmentation

3D part segmentation plays a crucial role in understanding both the compositional structure and functionality of 3D objects. To systematically evaluate existing part segmentation methods, especially class-agnostic ones, we establish a segmentation benchmark based on our PartNeXt dataset. Specifically, we select 5 objects per category, resulting in a total of 250 objects in the evaluation set. All leaf nodes in the annotation hierarchy are treated as ground truth labels, thereby enabling an

| Method | Category mIoU (%) | | | | | | | | | | mIoU |
|---|---|---|---|---|---|---|---|---|---|---|---|
| | Bed | Bottle | Chair | Knife | Table | Controller | Fan | Glasses | Monitor | Wrench | |
| SAMPart3D | 17.51 | 47.71 | 28.49 | 61.08 | 25.86 | 24.00 | 31.12 | 28.34 | 25.70 | 40.53 | 36.78 |
| SAMesh | **82.59** | 35.63 | **72.57** | 51.19 | **64.81** | **47.71** | **56.72** | 33.38 | 45.16 | 52.17 | **51.57** |
| PartField | 24.77 | **67.91** | 43.78 | **68.22** | 53.26 | 41.57 | 46.66 | **55.57** | **45.97** | **60.53** | 50.22 |

Table 2: Per-category IoU and mean IoU (mIoU) comparison across different methods. We present results for 10 representative categories: the first five are PartNet categories, while the latter five are novel categories beyond PartNet. Please refer to the appendix for segmentation results on more categories.

assessment of each method's ability to perform fine-grained segmentation. Following the evaluation protocol of PartField [25], we adopt mean Intersection-over-Union (mIoU) as our evaluation metric. For each ground truth part, we calculate the IoU with all predicted parts and record the maximum IoU achieved as the model's score for that part. These maximum IoU values are then averaged across all parts to obtain the final mIoU.

We evaluate recent SOTA part segmentation methods, including PartField[25], SAMPart3D[48], and SAMesh[39]. As shown in Figure 4 and Table 2, our analysis reveals distinct characteristics of each method: SAMesh demonstrates relatively strong performance in fine-grained segmentation but tends to produce over-segmented results. PartField occasionally fails to separate adjacent connected regions, while SAMPart3D struggles to maintain segmentation continuity in weakly-textured areas and exhibits inconsistent granularity control. Overall, current 3D part segmentation methods still face significant challenges in handling fine-grained segmentation tasks, particularly in balancing segmentation granularity with semantic consistency.

## 4.2 Part-Centric 3D Question Answering

To advance research on part-level reasoning in 3D vision, we propose a benchmark dedicated to evaluating model understanding of 3D object part structures. This benchmark is designed to assess a model's capability to understand, localize, and reason about fine-grained part structures in complex 3D objects. Specifically, we include three representative tasks: part counting, part classification, and part grounding. These tasks collectively probe a model's semantic granularity, structural awareness, and spatial correspondence in the context of 3D objects.

- **Part Counting:** Given a 3D object represented as a point cloud and a target part name, the model should predict the total number of target part instances in the object, testing object decomposition, with a downstream LLM converting text to numbers.

- **Part Classification:** Given ground-truth labels, points of a specific part are rendered in red, and the model should name the highlighted region, with a downstream LLM verifying correctness.

- **Part Grounding:** Given the full point cloud and a part query, the model should locate the part by predicting its bounding box corners.

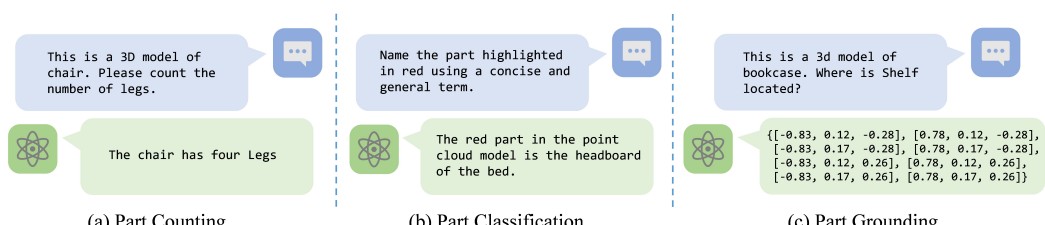

(a) Part Counting      (b) Part Classification      (c) Part Grounding

Figure 5: **Representative prompt–response pairs used to evaluate 3D part-level understanding.** (a) Part Counting: the model is requested to enumerate the number of legs in a chair. (b) Part Classification: the model must name the part highlighted in red within the point-cloud bed. (c) Part Grounding: the model is asked to localize the "Shelf" of a bookcase by outputting the eight corner coordinates of its bounding box.

We evaluate three existing multimodal language models adapted for 3D tasks: PointLLM [46], ShapeLLM [35], and 3DLLM [11]. The prompts used in our benchmark can be broadly categorized into two types: (1) category-aware prompts, where the category of the 3D object is provided to the model prior to querying; and (2) category-agnostic prompts, where the model is required to infer part-level semantics and structure without any prior knowledge of the object class.

Table 3 reports the performance of all three models across the above tasks. We observe that their performance on part counting and grounding remains limited. These results suggest that current 3D vision-language models lack fine-grained structural understanding and call for further research in this direction.

| | Part count(MAE) | | | Classification(Acc) | | Grounding(IoU) | | |
|---|---|---|---|---|---|---|---|---|
| | 3DLLM | PointLLM | ShapeLLM | PointLLM | ShapeLLM | 3DLLM | PointLLM | ShapeLLM |
| w category | 2.16 | 1.87 | 1.72 | 0.22 | 0.25 | \ | \ | 0.33 |
| wo category | 2.46 | 1.79 | 1.85 | 0.18 | 0.08 | \ | \ | 0.30 |

Table 3: Comparison of different 3D object part-level understanding models across three tasks. Note that '\' means fails to response reasonable boundingbox.

### 4.3 Analysis on 3D Promptable Segmentation

| Evaluation Dataset | Training Dataset | IoU@1 | IoU@3 | IoU@5 | IoU@7 | IoU@10 |
|---|---|---|---|---|---|---|
| PartNet-Mobility | PartNet | 39.0 | 53.7 | 58.6 | 60.9 | 62.9 |
| | PartNeXt | 40.2 | 57.5 | 63.2 | 65.0 | 67.4 |
| | Mixture | **40.4** | **58.3** | **64.1** | **66.9** | **68.7** |
| PartNeXt | PartNet | 39.9 | 53.9 | 58.4 | 60.4 | 60.3 |
| | PartNeXt | 44.3 | 60.1 | 63.2 | 64.8 | 65.9 |
| | Mixture | **45.3** | **61.7** | **65.3** | **66.6** | **67.6** |

Table 4: Comparison of Point-SAM [53] models trained on different datasets. The metric IoU@k is reported for 3D promptable segmentation, where $k$ denotes the number of prompt points.

To further demonstrate the quality of our dataset, we conduct experiments by training a high-capacity 3D promptable segmentation model, Point-SAM [53], on PartNeXt. For a comprehensive comparison, we train the model on three different dataset configurations: (1) PartNet, (2) PartNeXt, and (3) a combination of both. The setup is designed to evaluate the effect of the training dataset on the model's generalization performance. Following the evaluation protocol established in Point-SAM, we use the PartNet-Mobility dataset [44] to evaluate models on 3 categories (scissors, refrigerators, and doors). Additionally, to further assess zero-shot transfer, we also hold out 3 categories (scissors, microwave oven, and floor lamp) from the PartNeXt training data as the test set. All models are trained on 4 NVIDIA A6000 GPUs, with a batch size of 16 for a total of 50,000 optimization steps.

The results in Table 4 indicate a clear performance improvement when the model is trained on the mixture dataset. Notably, training on the PartNeXt dataset alone also yields significant gains. These findings validate the high quality of our part annotations and highlight the benefit of expanding category diversity to enhance model generalization. Such a richly annotated and diverse dataset provides a crucial foundation for improving the generalizability of future 3D foundation models.

## 5 Limitations and Conclusion

**Limitations** Currently, PartNeXt faces three main limitations. Firstly, to ensure high-quality annotations, PartNeXt currently includes only 23,519 models, but we are actively working on expanding the dataset by incorporating more data from ObjaverseXL [4]. Secondly, each category in PartNeXt requires a carefully predefined fine-grained part hierarchy, which limits our ability to annotate open-vocabulary datasets. We are exploring open-vocabulary part annotations through deeper integration with VLMs. Thirdly, PartNeXt currently provides only plain part name annotations

for each node. Introducing caption or physical attribute annotations for both category and part could greatly enrich the information of the PartNeXt dataset.

**Conclusion**    In this work, we present PartNeXt, a next-generation dataset tailored for fine-grained, hierarchically structured 3D part understanding. Leveraging a web-based annotation system guided by connectivity-aware part prompts, we efficiently annotate a large-scale collection of 23519 3D models spanning 50 diverse categories, with 350187 parts in total. By directly annotating on textured mesh, our dataset can also provide native texture for each part, which is an essential modality for comprehensive 3D understanding. The proposed class-agnostic segmentation and 3D part-centric QA benchmark enables a comprehensive evaluation of a model's understanding at the part level. Through our experiments, we observe that current 3D understanding models still exhibit significant limitations in understanding fine-grained part-level semantics. Therefore, we envision our proposed dataset, PartNeXt, as a high-quality foundational dataset to support the development of the next generation of part-level 3D understanding models.

# 6    Acknowledgement

This work was supported by Shanghai Pujiang Program (24PJA080), the National Natural Science Foundation of China under Grant W2431046, the MoE Key Lab of Intelligent Perception and Human-Machine Collaboration (ShanghaiTech University), the Shanghai Frontiers Science Center of Human-centered Artificial Intelligence, and the HPC Platform of ShanghaiTech University. We also appreciate Benyuan AI Data for providing support in data annotation, as well as the 35 annotators for their valuable contributions.

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

# A   Detail of Annotation Platform

We first provide additional details about the annotation platform we designed and used for large-scale data annotation. A video demonstrating our annotation workflow is included in the supplementary materials.

During the data annotation process, we found that some categories are difficult to distinguish using text similarity-based methods. Additionally, some models contain multiple objects. To address these issues, we merged some categories. Specifically, due to the limitations of Cap3D's multi-view captioning approach, it's difficult to ensure accurate captions for every viewpoint, which can result in inaccurate label. One of the most prominent cases we observed was in tool-related categories such as pliers, scissors, and hammers. Therefore, we merged some categories to allow annotators to select the correct category. For models containing multiple objects, we also grouped these into a single category to allow annotators to label all objects in the scene. Our merged categories include:

- **Tool**: axe, fork, hammer, pickaxe, pliers, scissors, screwdriver, shovel, spoon, wrench, and knife.
- **Liquid Container**: mug, bottle, cup.
- **Light**: chandelier, floor lamp, desk lamp.
- **Computer Device**: keyboard, mouse, monitor, laptop, controller, desk, chair, mug, desk lamp, headphones.

After completing the annotations, we decomposed these merged categories back into their original, individual categories. For models with multiple objects, we also split the original model to extract individual objects, thereby obtaining separate models.

# B   Dataset Statistic

Here we provide detail statistic of PartNeXt dataset, as shown in Table 5

| | All | Axe | Bag | Bed | Bookcase | Bottle | Buck | Cam | Chair | Chandeller | Mouse | Control | Cup | Door | Fan | Flashlight | F-Lamp |
|---|---|---|---|---|---|---|---|---|---|---|---|---|---|---|---|---|---|
| #S | 23519 | 1628 | 69 | 1454 | 574 | 1204 | 168 | 8 | 4277 | 1355 | 129 | 78 | 361 | 9 | 93 | 118 | 78 |
| #P | 350187 | 7142 | 589 | 33661 | 15018 | 4196 | 1974 | 81 | 60819 | 51678 | 520 | 1640 | 1056 | 48 | 950 | 715 | 718 |
| $P_{Med}$ | 15 | 7 | 8 | 27 | 24 | 6 | 12 | 15 | 19 | 24 | 8 | 24 | 6 | 9 | 15 | 10 | 14 |
| $D_{Med}$ | 4 | 3 | 3 | 8 | 4 | 3 | 4 | 4 | 4 | 4 | 5 | 4 | 3 | 5 | 5 | 3 | 4 |
| $D_{Max}$ | 9 | 3 | 3 | 9 | 5 | 3 | 5 | 5 | 6 | 5 | 5 | 4 | 3 | 5 | 5 | 3 | 4 |

| | Fork | Glass | Guitar | Ham. | H-phone | Keyboard | Knife | Lamp | Lap. | Micro. | Monitor | Mug | P-lock | Pen | Piano | Pickaxe |
|---|---|---|---|---|---|---|---|---|---|---|---|---|---|---|---|---|
| #S | 76 | 291 | 279 | 583 | 136 | 228 | 998 | 950 | 94 | 67 | 332 | 498 | 18 | 146 | 106 | 124 |
| #P | 487 | 2833 | 9574 | 2843 | 1225 | 22045 | 3815 | 11073 | 9385 | 1042 | 1941 | 1133 | 83 | 864 | 6945 | 560 |
| $P_{Med}$ | 11 | 17 | 46 | 7 | 10 | 88 | 6 | 13 | 82 | 15 | 7 | 4 | 9 | 10 | 68 | 7 |
| $D_{Med}$ | 4 | 5 | 5 | 3 | 3 | 4 | 3 | 4 | 4 | 4 | 3 | 3 | 5 | 5 | 4 | 3 |
| $D_{Max}$ | 7 | 5 | 5 | 3 | 3 | 4 | 3 | 4 | 4 | 4 | 3 | 3 | 5 | 5 | 5 | 3 |

| | Plier | Sciss. | Screw | Shovel | Skate. | Sofa | Spoon | S-Ladder | Sword | Table | Teapot | Toast. | Toilet | Umb. | Wash. | Watch | WiFi | Wrench |
|---|---|---|---|---|---|---|---|---|---|---|---|---|---|---|---|---|---|---|
| #S | 65 | 40 | 71 | 73 | 141 | 3139 | 101 | 22 | 151 | 2326 | 348 | 28 | 137 | 65 | 19 | 115 | 8 | 141 |
| #P | 523 | 229 | 218 | 525 | 2461 | 48135 | 285 | 433 | 647 | 31836 | 3180 | 549 | 1133 | 1111 | 280 | 1256 | 123 | 610 |
| $P_{Med}$ | 14 | 9 | 5 | 13 | 25 | 18 | 5 | 17 | 6 | 14 | 12 | 20 | 12 | 19 | 21 | 19 | 23 | 7 |
| $D_{Med}$ | 4 | 4 | 3 | 6 | 6 | 3 | 3 | 3 | 3 | 5 | 3 | 4 | 4 | 5 | 4 | 6 | 4 | 4 |
| $D_{Max}$ | 4 | 4 | 3 | 6 | 6 | 4 | 3 | 4 | 3 | 8 | 3 | 6 | 4 | 6 | 4 | 7 | 4 | 5 |

Table 5: **PartNeXt Dataset Statistic**. #S represent number of annotated objects, #P as the number of total annotated parts, $P_{Med}$ is median number of parts, $D_{Med}$ is median number of hierarchy depth, $D_{Max}$ is maximum number of hierarchy depth.

Here we provide the visualization of our predefined hierarchy and more annotations. As shown in Figure 10, Figure 11, Figure 12, Figure 13.

Although we provide annotators with fine-grained hierarchies, we cannot guarantee that the hierarchy covers all possible parts. Therefore, during annotation, annotators are allowed to label parts that are not included in the hierarchy as "Other". We also provide statistics on the "Other" parts, as shown in Table 6.

| Number of Other | 0 | 1 | 2 | 3 | 4 | 5 | >5 |
|---|---|---|---|---|---|---|---|
| Percentage % | 82.22 | 13.82 | 1.77 | 0.30 | 0.38 | 0.20 | 1.30 |

Table 6: **Distribution of the Number of Other Instances.** The table shows the percentage of occurrences for each count category.

## C  Comparison with Recent 3D Part Dataset

We present a comparison between PartNeXt and recent 3D part-level datasets, as shown in Table 7. This includes detailed information on several 3D object part-level datasets such as PartNet[32] and PartObjaverseTiny from SAMPart3D[48].

| Dataset | Texture | Raw Geometry | Semantic | Hierarchy | # Shape | # Category |
|---|---|---|---|---|---|---|
| PartNet | × | × | ✓ | ✓ | 26,671 | 24 |
| PartObjaverseTiny | ✓ | ✓ | ✓ | × | 200 | 8 |
| PartNeXt | ✓ | ✓ | ✓ | ✓ | 23,519 | 50 |

Table 7: **Comparison between PartNeXt and recent 3D Part Datasets.** Raw Geometry indicates if the part remains the raw geometry (PartNet lacks texture and raw geometry due to remesh operation).

The PartNet[32] dataset is annotated directly on textureless models and requires remeshing during the annotation process. As a result, the final annotations lack both texture and original geometry, omitting important color information and potential geometric details (such as mesh topology and face distribution), which are crucial for comprehensive 3D understanding. Besides, PartNet requires manually drawn cutting lines after remeshing to achieve segmentation. As a result, the boundaries of the parts are often not smooth. In contrast, our method leverages per-face annotations, effectively avoiding these issues, the comparison is shown in Figure 6.

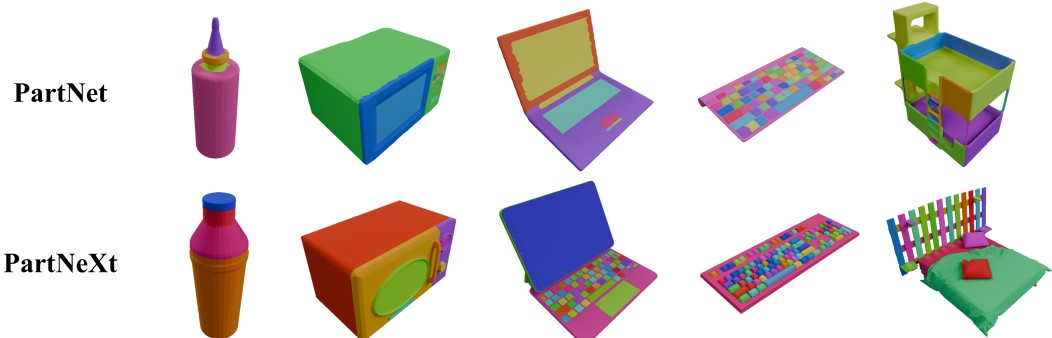

Figure 6: **Visualization of PartNet and PartNeXt results**. Since PartNet uses remeshing to obtain finer-grained parts, the mesh undergoes deformation, lacks texture, and requires manually drawn cutting lines after remeshing to achieve segmentation. As a result, the boundaries of the parts are often not smooth.

PartObjaverseTiny[48] retains the original geometry during annotation, but it lacks a hierarchical structure, making it unsuitable for models that require multi-granularity segmentation (e.g., PointSAM[53]).

In terms of scale, our dataset is comparable to PartNet[32] in quantity, whereas PartObjaverseTiny[48] contains only 200 models—insufficient for large-scale 3D applications. As for category diversity, our dataset includes more than twice the number of categories compared to PartNet[32]. Although PartObjaverseTiny[48] claims to have subcategories under its eight main categories, the limited quantity constrains its scalability for broader 3D model tasks.

In summary, PartNeXt offers clear advantages in terms of dataset scale and category diversity. It also preserves complete texture and original geometric structure, along with hierarchical annotations, enabling more nuanced and semantically rich 3D understanding tasks. We believe that PartNeXt will provide a stronger foundation and broader potential for future research in 3D segmentation and understanding.

## D   Detail of Segmentation BenchMark

We provide the detailed evaluation result and settings for our proposed benchmark on class-agnostic 3D part instance segmentation. All evaluations were conducted on a server equipped with 2 Intel(R) Xeon(R) Platinum 8581C processor and 4 NVIDIA L40 GPUs across experiments for fairness.

As shown in Table 8, we report the per-category metrics for each method included in the benchmark.

We evaluate each method using its publicly available codebase, and follow their original settings and hypermeters. SAMesh [39] utilizes textureless meshes, PartField [25] uses colorless point clouds, sampled from mesh surfaces, SAMPart3D [48] requires point clouds with color and normal attributes, so we sample colorized point clouds from textured meshes as input. Following the evaluation code of PartField, we convert the predicted masks from all methods into a unified format before computing the mIoU.

| Method | Category mIoU (%) | | | | | | | | | |
|---|---|---|---|---|---|---|---|---|---|---|
| | Axe | Bed | Bookcase | Bottle | Bucket | Camera | Chair | Chandelier | Com_Keyboard | Com_Mouse |
| SAMPart3D | 61.53 | 17.51 | 16.78 | 47.71 | 28.08 | 27.34 | 28.49 | 17.26 | 32.52 | 28.28 |
| SAMesh | 71.50 | 82.59 | 39.30 | 35.63 | 60.45 | 39.57 | 72.57 | 37.05 | 89.98 | 45.77 |
| PartField | 65.60 | 24.77 | 26.68 | 67.91 | 31.76 | 50.05 | 43.78 | 26.48 | 27.14 | 52.05 |
| | Controller | Cup | Door | Fan | Flashlight | Floor_lamp | Fork | Glasses | Guitar | Hammer |
| | 24.00 | 55.06 | 42.76 | 31.12 | 37.02 | 36.84 | 59.96 | 28.34 | 6.75 | 60.74 |
| | 47.71 | 37.78 | 64.66 | 56.72 | 32.79 | 59.87 | 36.26 | 33.38 | 33.68 | 43.33 |
| | 41.57 | 72.90 | 63.48 | 46.66 | 45.11 | 70.59 | 66.03 | 55.57 | 23.13 | 64.58 |
| | Bag | Headphone | Knife | Lamp | Laptop | Microwave | Monitor | Mug | Padlock | Pen |
| | 42.60 | 28.33 | 61.08 | 36.60 | 6.92 | 18.29 | 25.70 | 48.70 | 60.35 | 40.60 |
| | 68.69 | 45.41 | 51.19 | 58.49 | 24.08 | 42.16 | 45.16 | 82.25 | 78.12 | 33.48 |
| | 52.88 | 44.08 | 68.22 | 53.32 | 12.34 | 38.48 | 45.97 | 75.37 | 68.85 | 56.83 |
| | Pickaxe | Piano | Pliers | Screwdriver | Scissors | Shovel | Skateboard | Sofa | Spoon | Step_ladder |
| | 57.63 | 23.10 | 36.27 | 68.72 | 47.11 | 35.35 | 23.51 | 23.03 | 68.26 | 50.26 |
| | 66.94 | 39.10 | 36.63 | 35.47 | 55.96 | 38.03 | 43.45 | 54.48 | 44.13 | 81.22 |
| | 68.48 | 19.51 | 50.90 | 61.78 | 63.90 | 50.91 | 42.44 | 55.01 | 62.18 | 52.29 |
| | Sword | Table | Teapot | Toaster | Toilet | Umbrella | Watch | Wrench | Washing_machine | Wifi_routers |
| | 59.87 | 25.86 | 43.36 | 25.49 | 29.70 | 32.58 | 27.65 | 40.53 | 17.87 | 25.86 |
| | 63.72 | 64.81 | 73.24 | 64.26 | 45.41 | 47.01 | 50.46 | 52.17 | 41.82 | 32.75 |
| | 51.85 | 53.26 | 73.83 | 46.98 | 38.46 | 38.45 | 31.29 | 52.94 | 39.88 | 60.53 |

Table 8: Full quantitative comparison on our segmentation benchmark.

## E   Detail of LLM BenchMark

We provide detailed evaluation settings for our proposed Part-Centric 3D Question Answering BenchMark.

This benchmark experiments are conducted on the same hardware setup: a server equipped with 2 Intel(R) Xeon(R) Platinum 8581C processor and 4 NVIDIA L40 GPUs.

Since the models we evaluate require point cloud inputs, we uniformly sample 8192 points from the mesh using area-based sampling via the Trimesh library to ensure fairness in evaluation.

In all experiments, we use publicly available code. However, we observe that current LLM for 3D understanding shows limited capabilities in instruction following and response formatting. Even with carefully designed prompts, these models struggle to generate well-formatted outputs. To address this issue, we utilize a language model for output postprocess. We deploy the Qwen3-14B[47] AWQ quantized model using the vLLM framework[16] on 4 NVIDIA GeForce 2080Ti GPUs for postprocess.

For the Part Count task, the outputs are converted into a single numerical value, the prompt is shown in Listing 1. For the Part Classification task, we directly use the LLM to verify the correctness of the prediction with predicted label and ground truth label as input, output is either "True" or "False". The prompt is shown in Listing 2

Listing 1: System prompt for part count output conversion.

```
system_prompt = """
    You are a precise information extraction system. Analyze
        the given sentence describing the quantity of
        components and output a JSON object containing the
        numerical value.
    Follow these rules:

    Extract only the exact numerical quantity mentioned (either
         as digits or words)
    Ignore non-quantitative descriptors like size/color/
        material
    Return 0 if no quantity is specified
    Use this format: {"number": <extracted_value>}

    Examples:
    Input: "The chair contains three cushions"
    Output: {"number": 3}

    Input: "There are 8 buttons on the device"
    Output: {"number": 8}

    Input: "The lamp comes with a bulb"
    Output: {"number": 1}

    Input: "No additional parts included"
    Output: {"number": 0}
"""
```

Listing 2: System prompt for part classification correctness verification.

```
system_prompt = """
    You are a precise semantic verification system.
    Given a sentence describing a part of an object and a
        ground truth label, determine whether the described
        part corresponds to the given label.

    Follow these rules:

    - Focus on the identity or function of the described part.
    - Determine whether the described part refers to the same
        concept as the ground truth label.
    - Ignore attributes like color, position, size, material
        unless they are essential to identify the part.
    - Do not require exact word match - semantic equivalence is
         acceptable (e.g., "display" matches "Screen").
    - Output only a JSON object in this format: {"output": "
        True"} or {"output": "False"}.
```

```
      - Do not include any other text.

    Examples:

    Input:
    Sentence: "The red-highlighted section is the soft part
        people rest their heads on while sleeping."
    Ground truth: "Pillow"
    Output:
    {"output": "True"}

    Input:
    Sentence: "The red-highlighted component is the handle used
        to carry the toolbox."
    Ground truth: "Handle"
    Output:
    {"output": "True"}

    Input:
    Sentence: "The highlighted area is the internal CPU used
        for processing data."
    Ground truth: "Pillow"
    Output:
    {"output": "False"}

    Input:
    Sentence: "The indicated region is a curved support that
        connects both lenses at the nose."
    Ground truth: "Temple"
    Output:
    {"output": "False"}

"""
```

## E.1 Detail of Part Count Task

For the Part Count task, we detail how prompts are provided to the model. We first select a set of parts from the hierarchies of each object category that are most meaningful for this task, for example, keys on a keyboard, fan blades of a fan, legs of a table or chair, and arms of a chandelier. The selected parts are typically repeated within a single object, and their counts vary across different object instances.

After processing with a 3D understanding LLM and a separate language model for output post-processing, we obtain a predicted part count for each specified part of every object. We evaluated the performance of different 3D understanding LLMs using **Mean Absolute Error** (MAE) as the evaluation metric. The calculation is as follows:

Given an object $i$ with $n_i$ parts that have predicted and ground-truth counts, the object-level MAE is defined as:

$$\text{MAE}_i = \frac{1}{n_i} \sum_{j=1}^{n_i} |\hat{c}_{i,j} - c_{i,j}| \tag{1}$$

where $\hat{c}_{i,j}$ and $c_{i,j}$ denote the predicted and ground-truth count of the $j$-th part in object $i$, respectively.

The overall MAE for the 3D understanding LLM is then computed by averaging over all $M$ objects:

$$\text{MAE}_{\text{avg}} = \frac{1}{M} \sum_{i=1}^{M} \text{MAE}_i \tag{2}$$

## E.2 Detail of Part Classification Task

For the Part Classification task, we detail how prompts are provided to the model. Since our dataset includes segmentation masks, when querying the LLM about a specific part, we highlight the corresponding point cloud segment in red to serve as an effective 3D prompt. Note that, as some objects may inherently contain red regions, we intentionally avoid selecting objects with large red areas to prevent ambiguity. Examples of our processed point cloud prompts are shown in Figure 7.

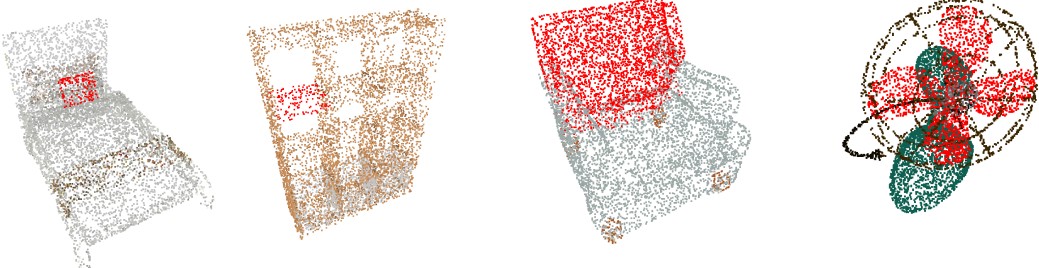

Figure 7: Visualization of Prompted Input for Part Classification Task, the target part is highlighted in red as 3d prompt.

## E.3 Detail of Part Grounding Task

For ShapeLLM[35], we follow the original evaluation code, requiring the model to output the coordinates of the eight vertices of the bounding box. We then compute the minimum and maximum values to calculate the IoU. An example of the output from ShapeLLM[35], along with the ground truth, is shown in Figure 8. These results indicate that current 3D language models are still in the early stages when it comes to the Part Grounding task.

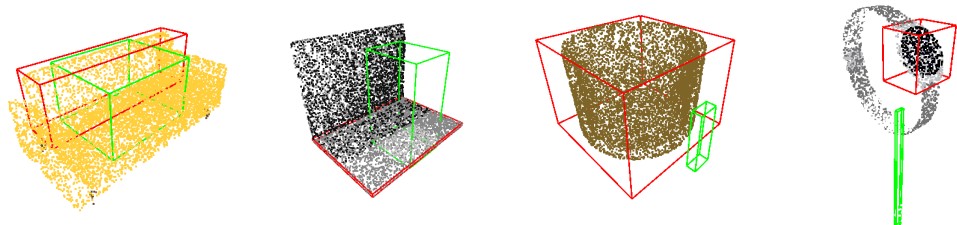

Figure 8: Visualization of ShapeLLM grounding result, the red box is the ground truth, while the green box is predict result.

In Figure 9, we present several output examples from 3DLLM[11] and PointLLM[46]. As shown, both types of models struggle to generate precise bounding box coordinates, indicates the weak ability of part grounding.

**Prompt 1**

```
Where is the screen?
```

**Output 1**

```
The screen is located in the middle of the computer setup.
```

**Prompt 2**

```
This is a monitor. Where is the screen?
```

**Output 2**

```
The screen is turned off, hence black.
```

**Prompt 3**

```
Where is the screen? Please answer with the coordinates of the
    eight vertices of its bounding box.
```

**Output 3**

```
The screen is at the center of the bounding box. The vertices
    are (0,0), (0,1), (1,1), and (1,0).
```

**Prompt 4**

```
This is a monitor. Where is the screen? Please answer with the
    coordinates of the eight vertices of its bounding box.
```

**Output 4**

```
The screen is located at the center of the monitor.
```

**Prompt 5**

```
Where is the screen? Please answer with the coordinates of the
    eight vertices of its bounding box, in the format: [[x1, y1
    , z1], [x2, y2, z2], ..., [x8, y8, z8]].
```

**Output 5**

```
The screen is located at the origin of the bounding box, which
    is also its center. The eight vertices of the box are: [[0,
    0, 0], [0, 0, 1], [0, 1, 0], [0, 1, 1], [1, 0, 0], [1, 0,
    1], [1, 1, 0], [1, 1, 1].
```

**Prompt 6**

```
This is a monitor. Where is the screen? Please answer with the
    coordinates of the eight vertices of its bounding box, in
    the format: [[x1, y1, z1], [x2, y2, z2], ..., [x8, y8, z8
    ]].
```

**Output 6**

```
The screen is at the center of the monitor.
```

Figure 9: Example response of 3DLLM and PointLLM under different prompts.

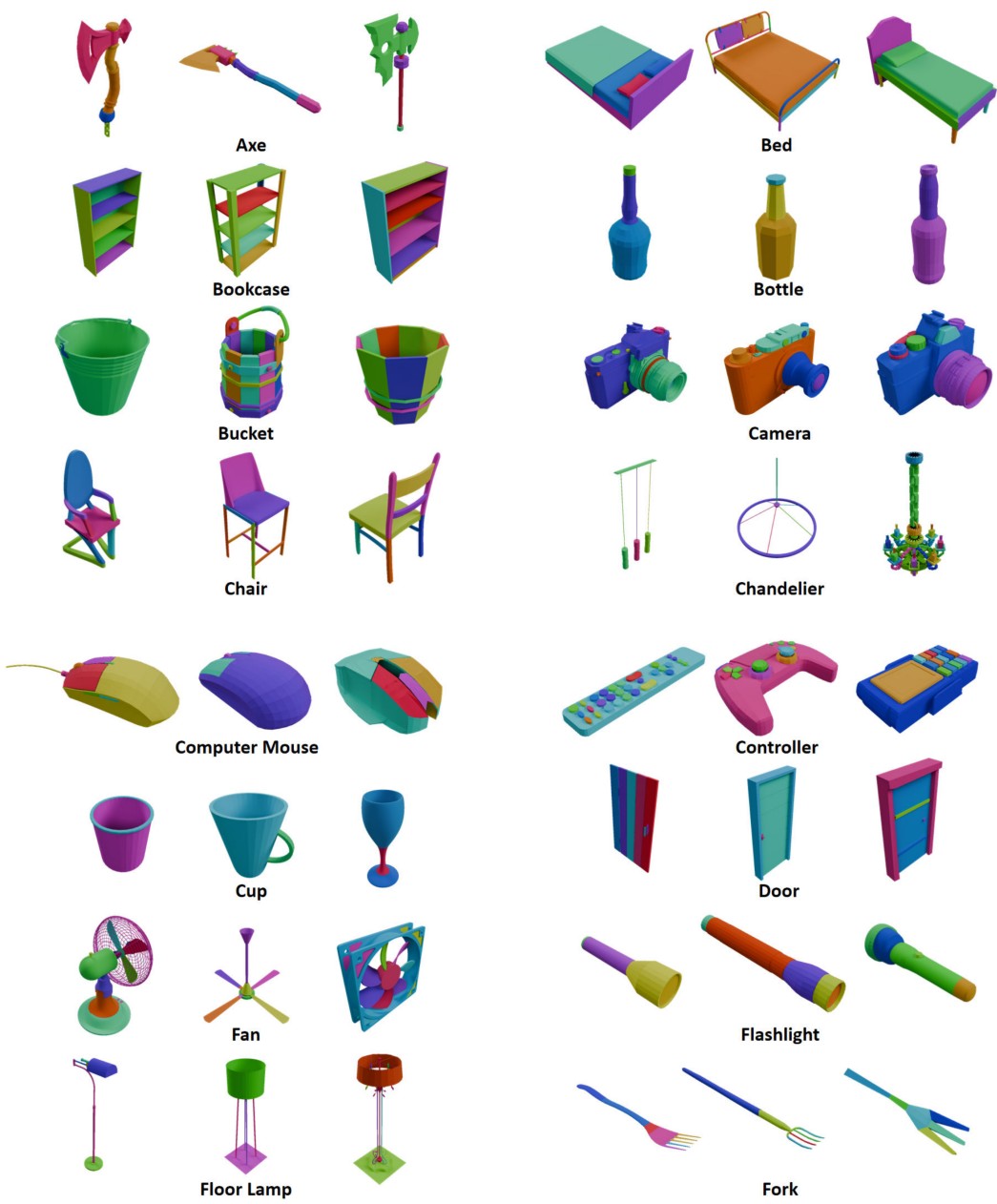

Figure 10: Visualization of our Annotation Results.

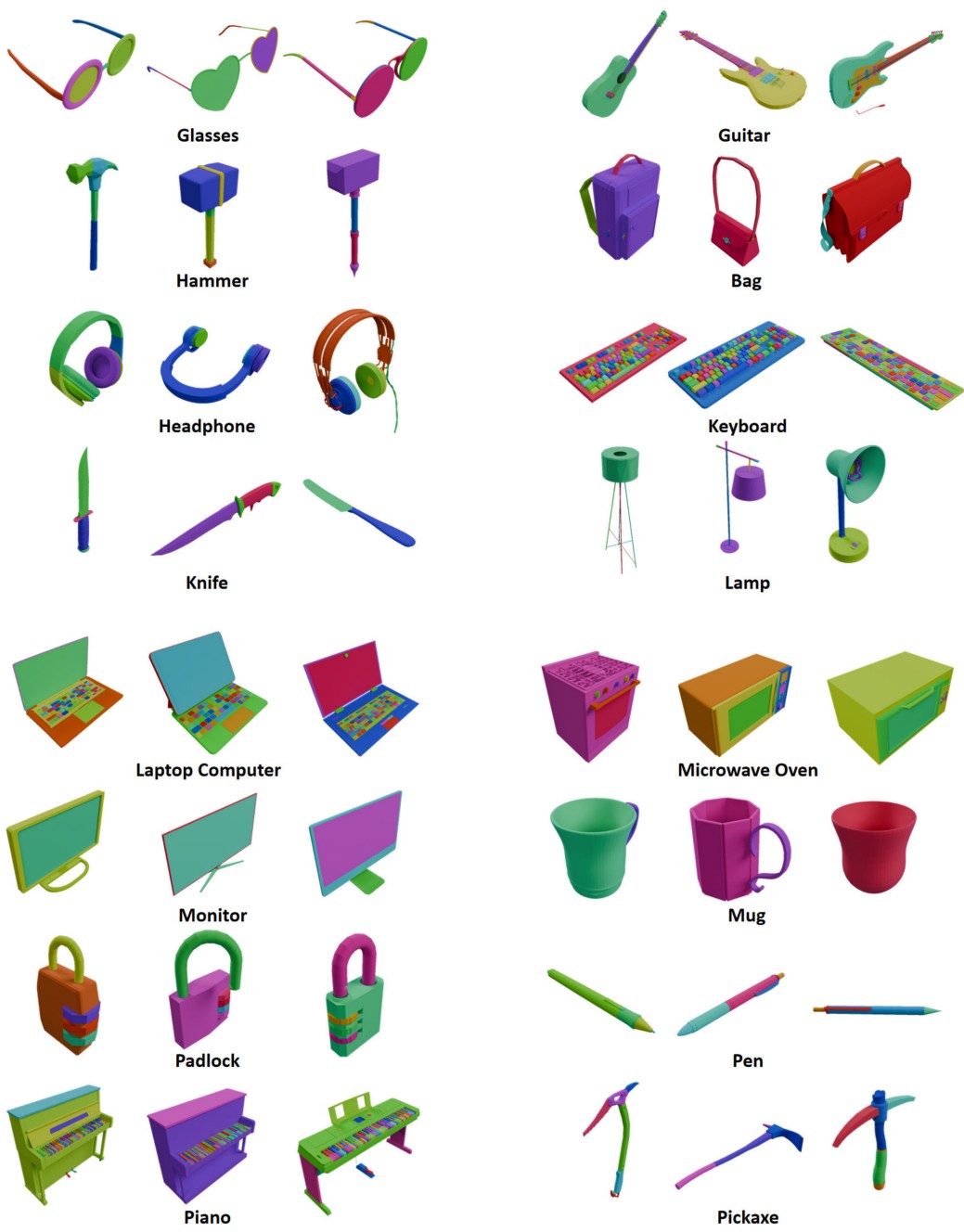

Figure 11: Visualization of our Annotation Results.

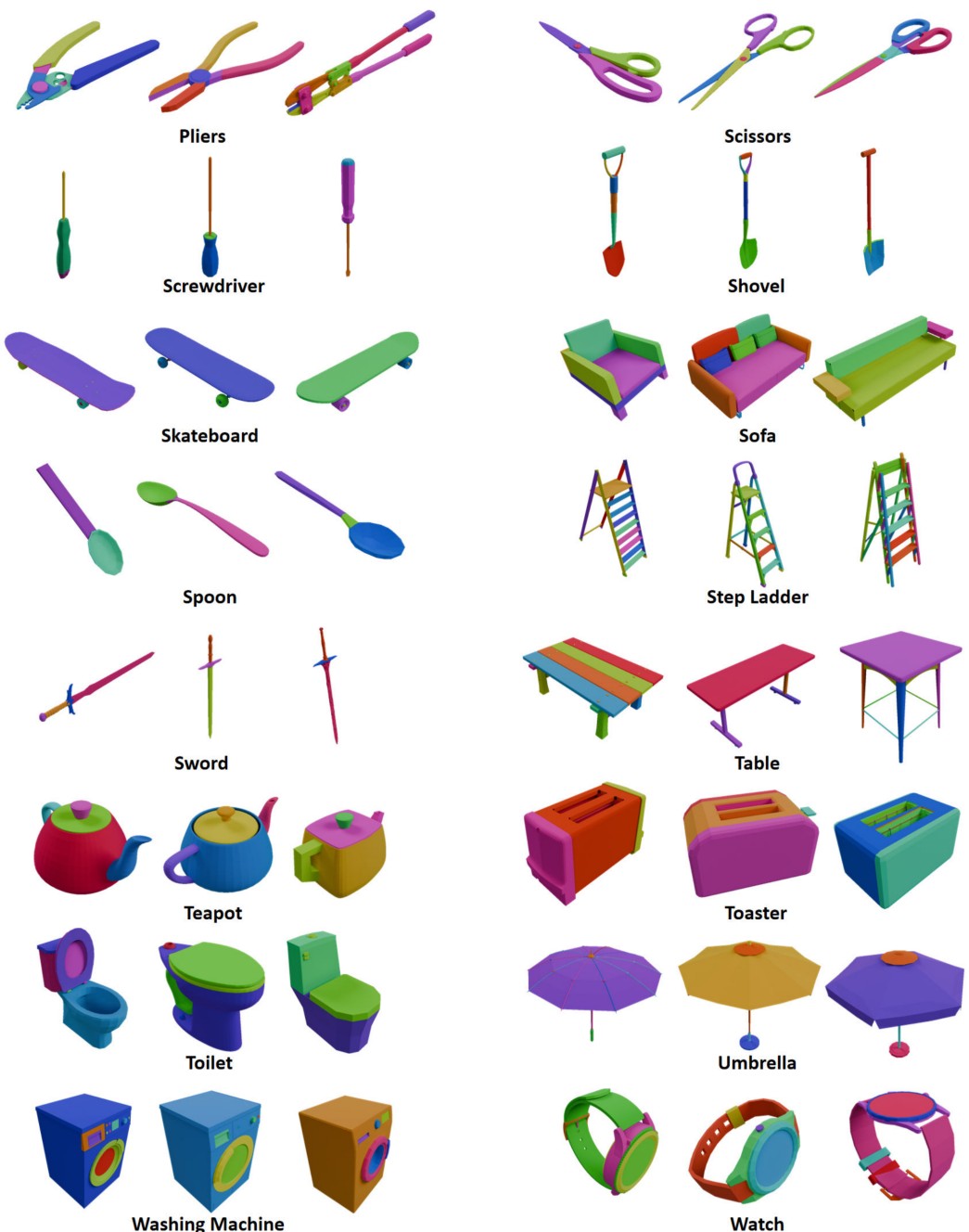

Figure 12: Visualization of our Annotation Results.

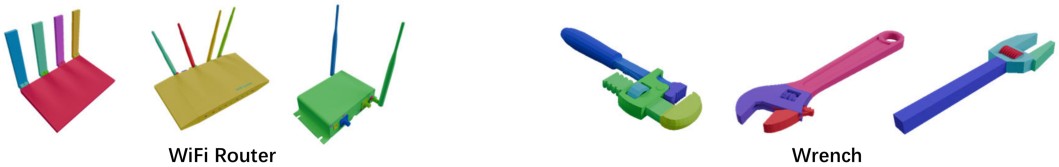

WiFi Router                                    Wrench

Figure 13: Visualization of our Annotation Results.

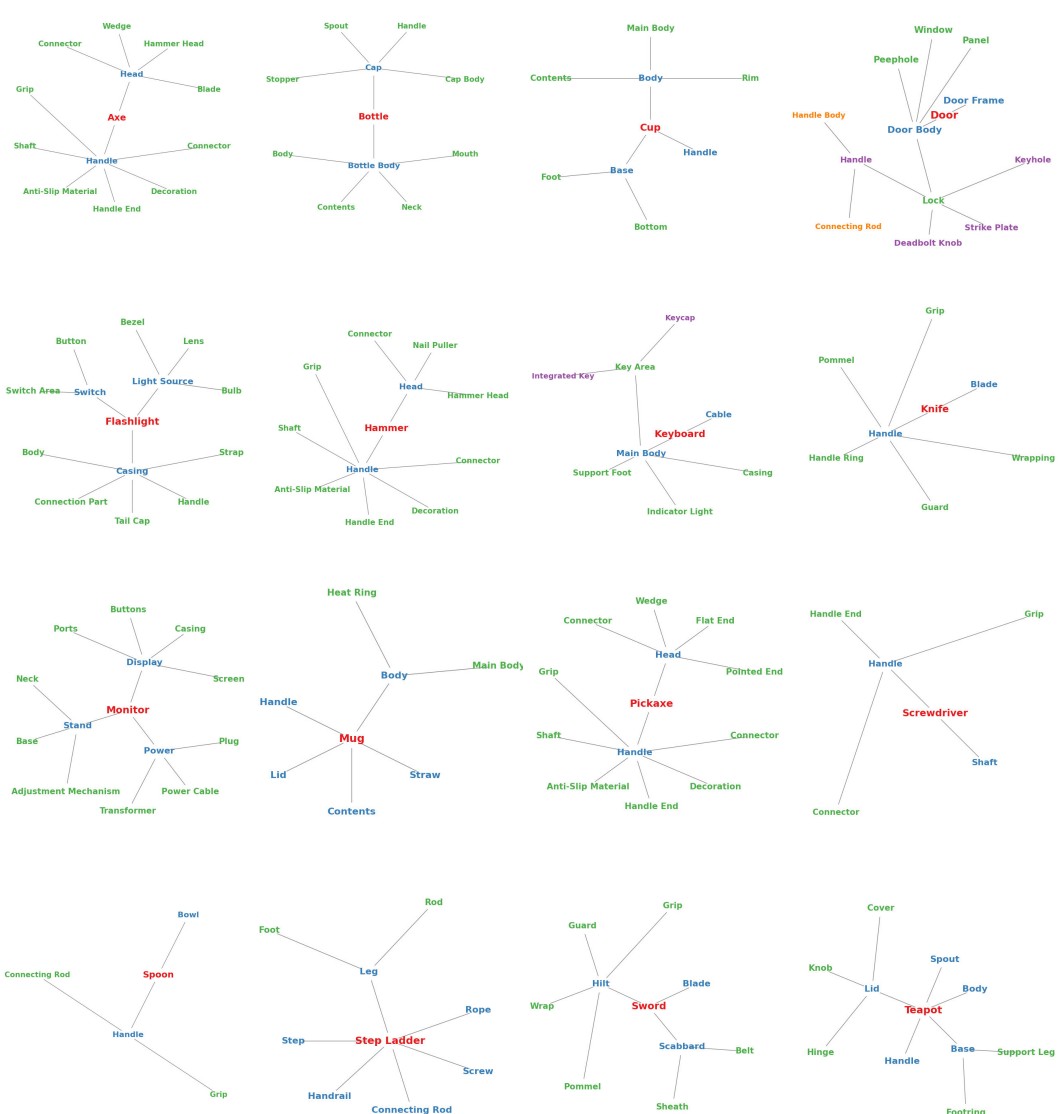

Figure 14: Visualization of our Predefined Hierarchy

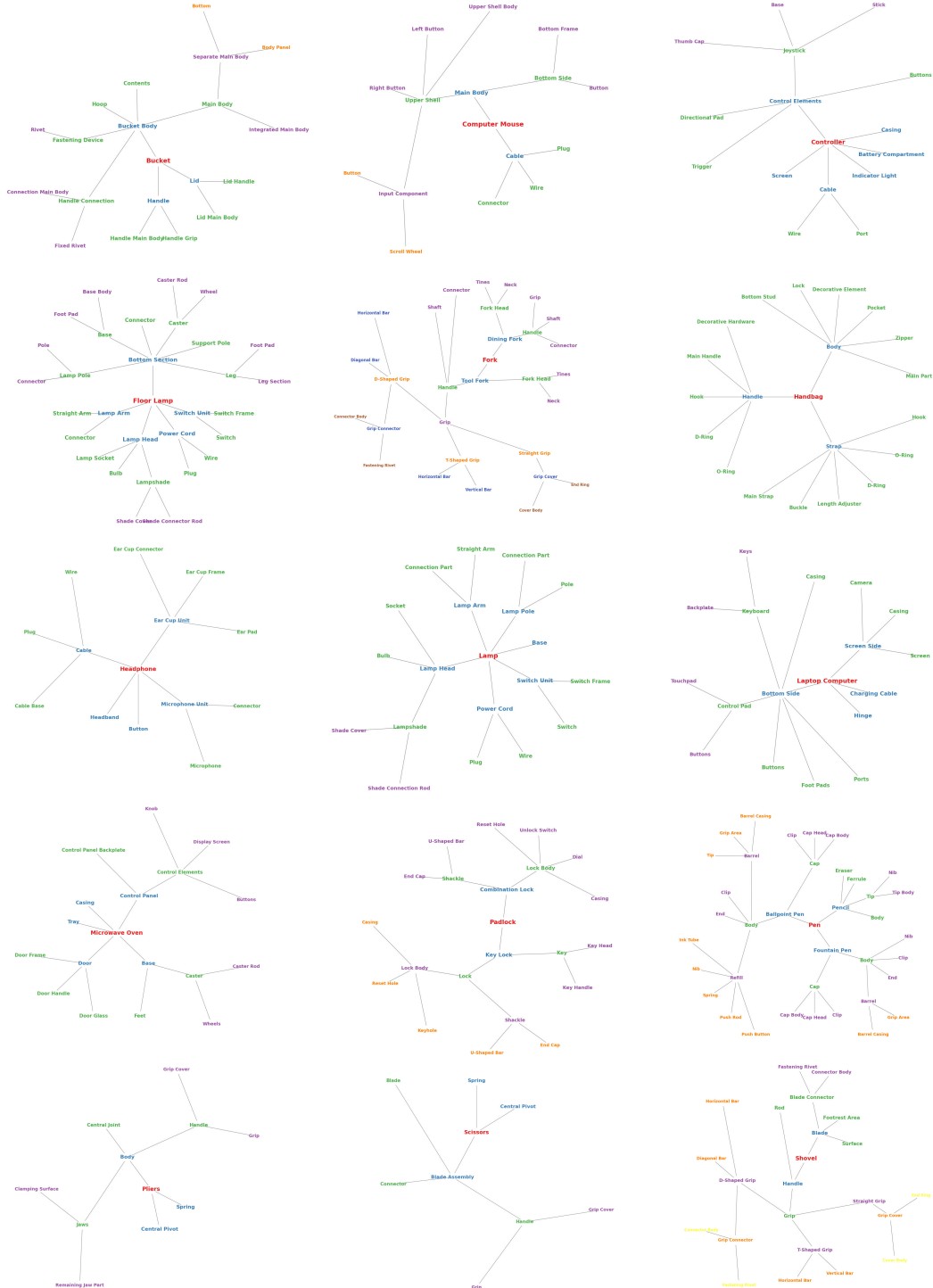

Figure 15: Visualization of our Predefined Hierarchy

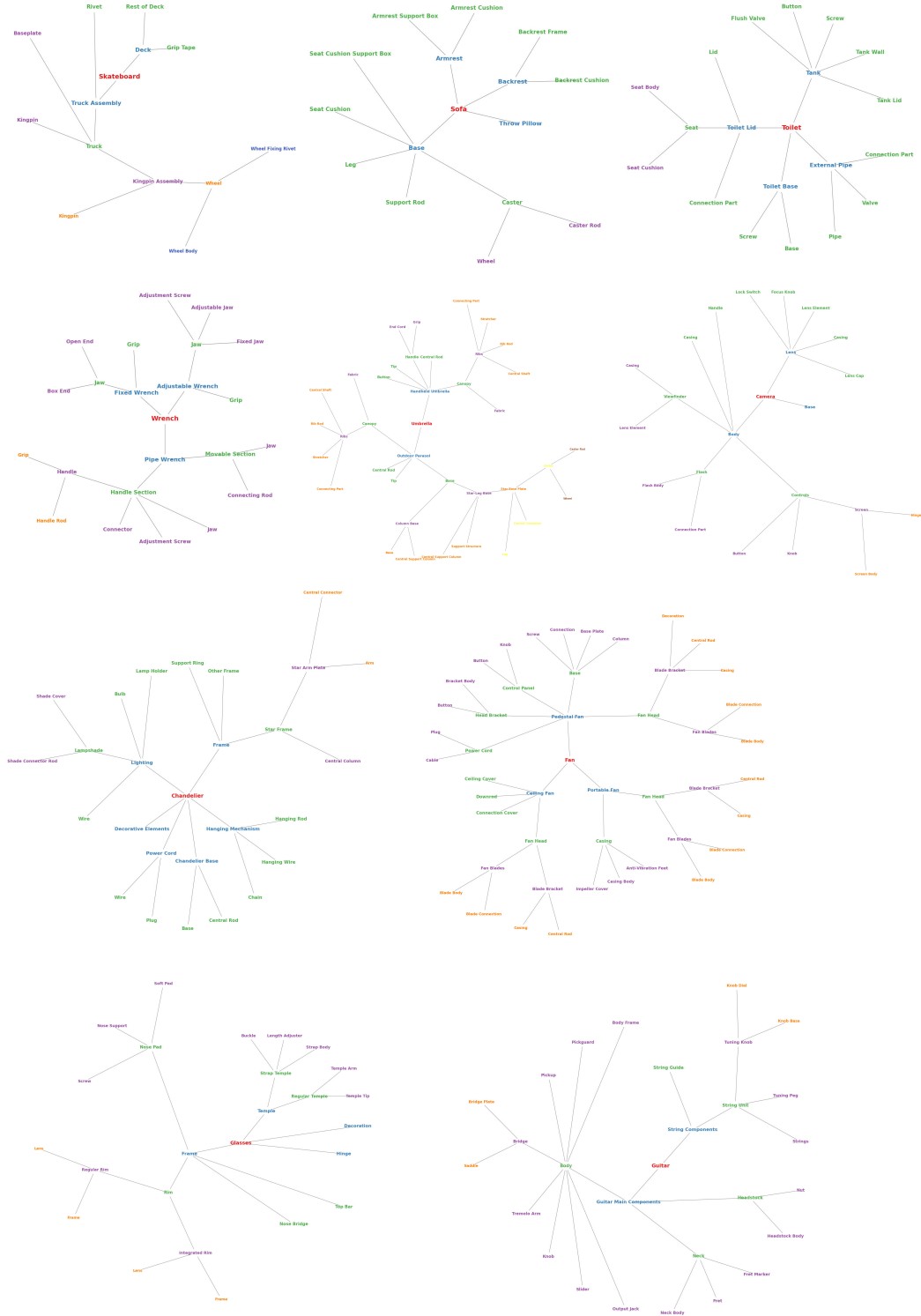

Figure 16: Visualization of our Predefined Hierarchy

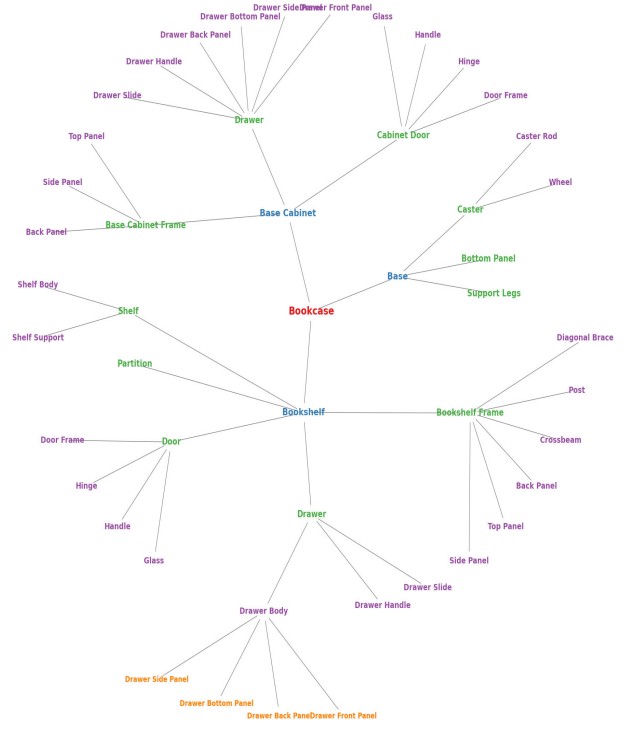

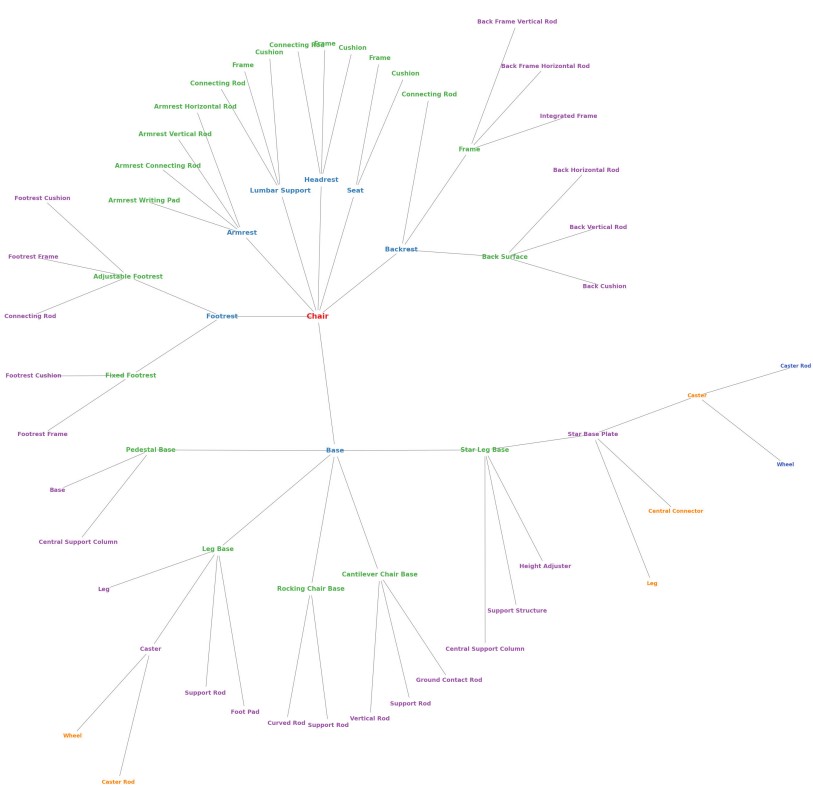

Figure 17: Visualization of our Predefined Hierarchy

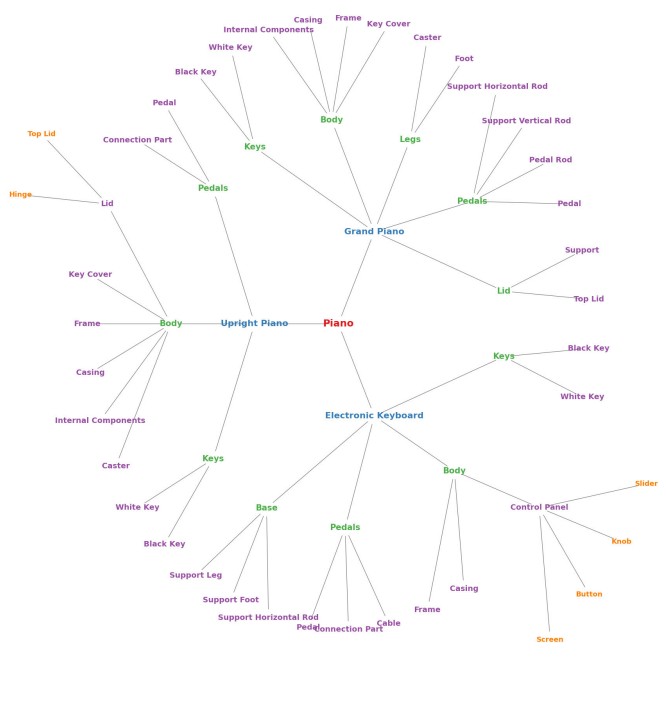

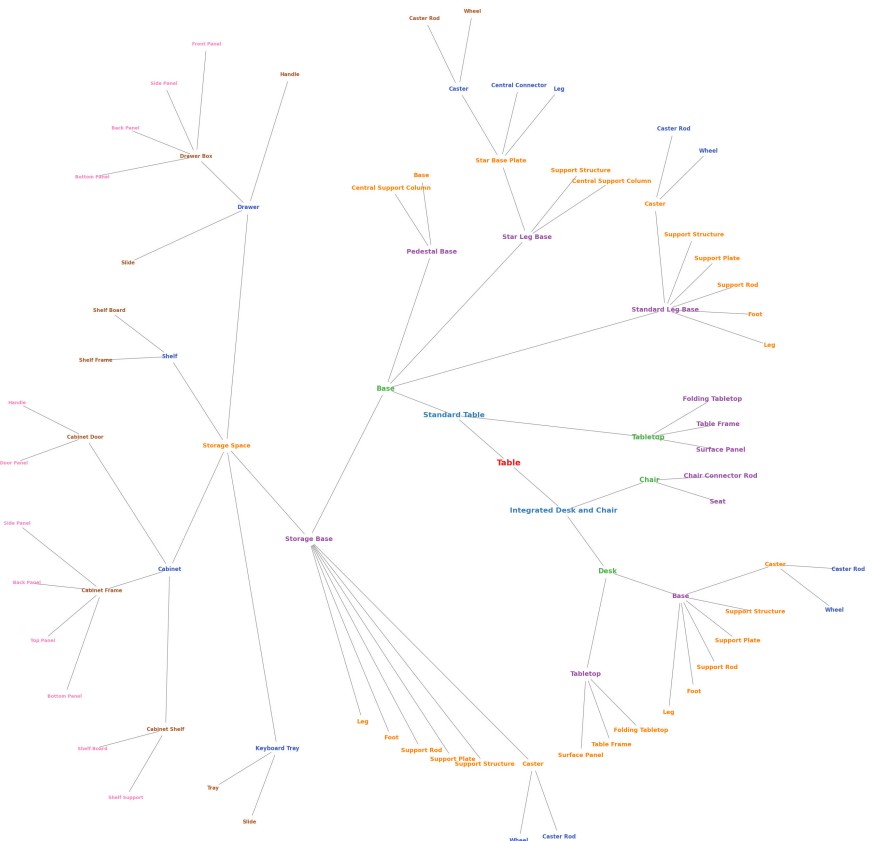

Figure 18: Visualization of our Predefined Hierarchy

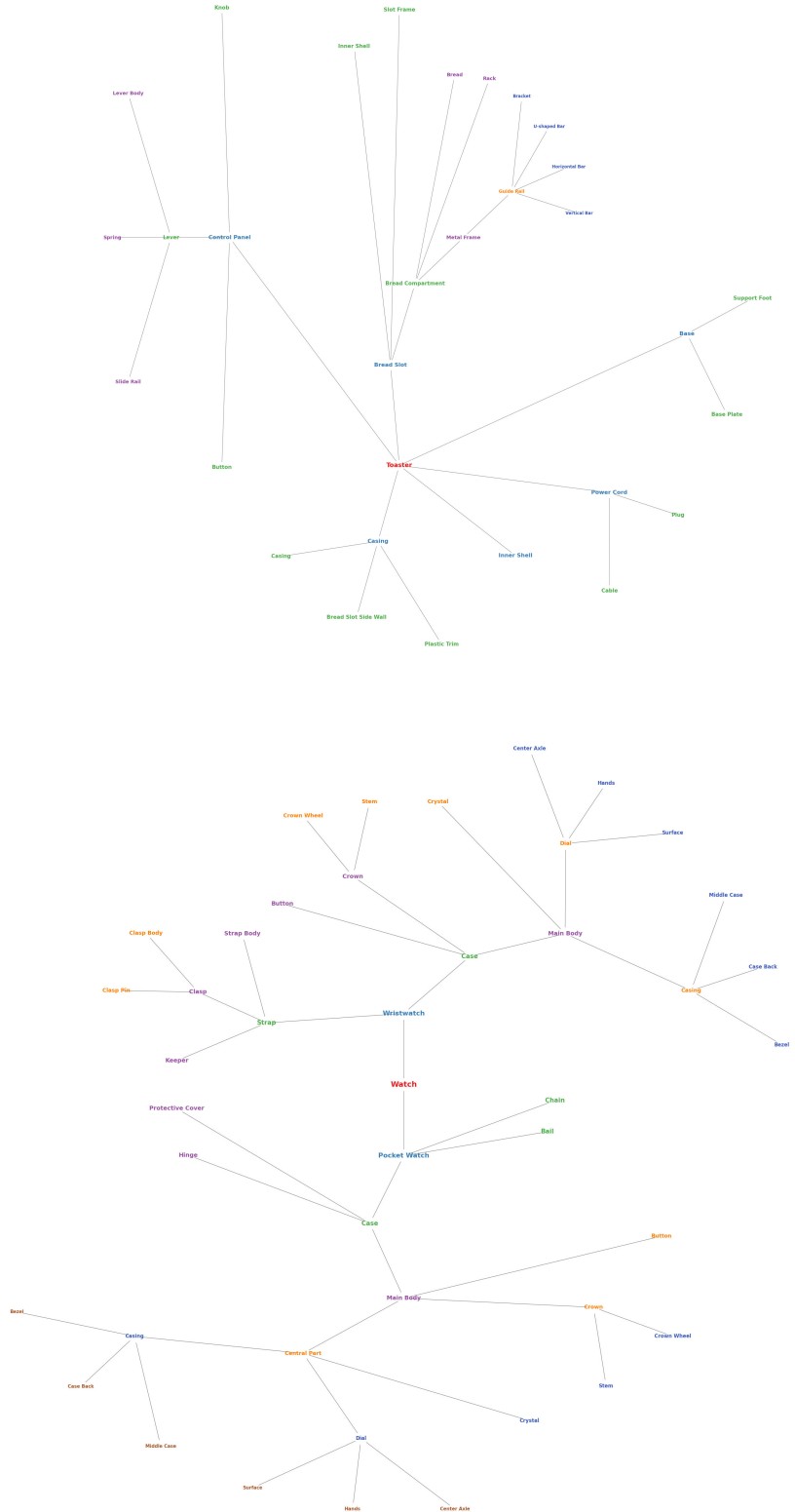

Figure 19: Visualization of our Predefined Hierarchy

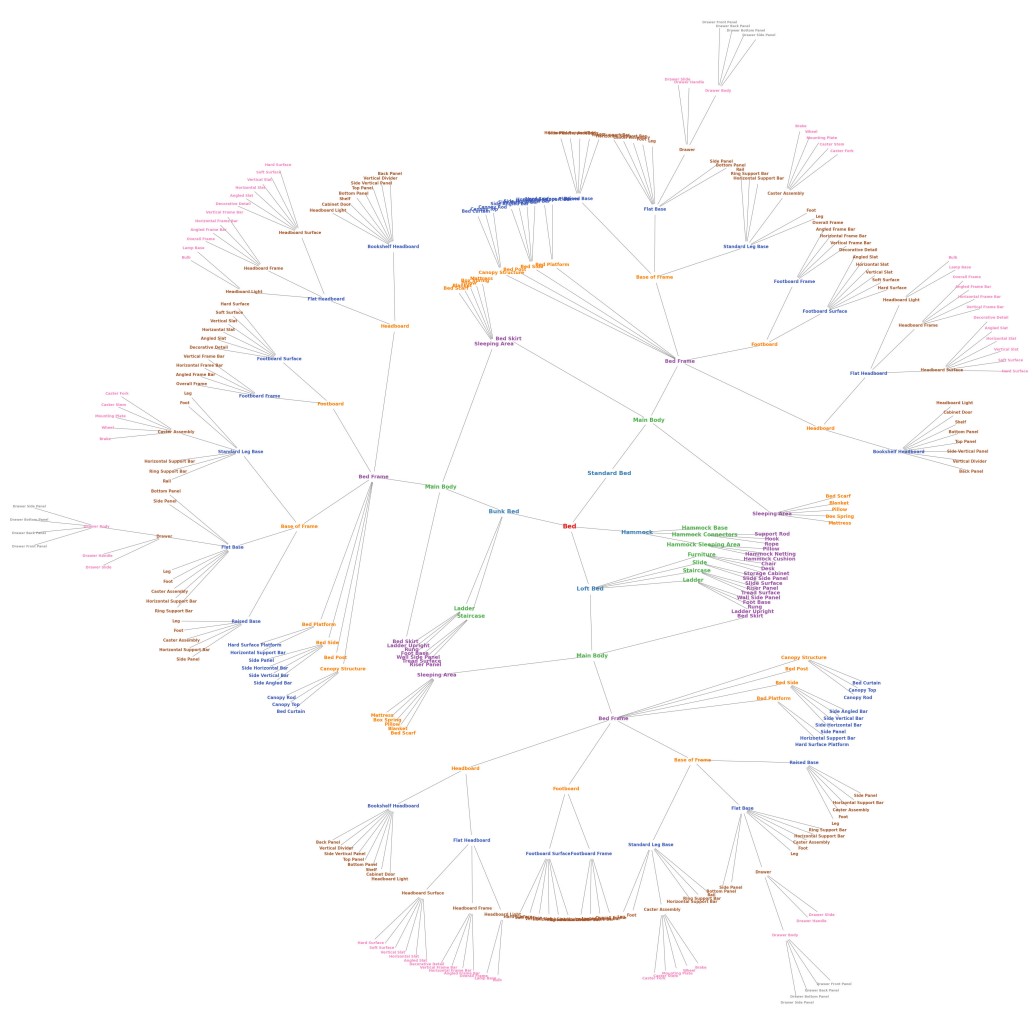

Figure 20: Visualization of our Predefined Hierarchy

