# OpenReview forum: "PartNeXt: A Next-Generation Dataset for Fine-Grained and Hierarchical 3D Part Understanding"
_NeurIPS.cc/2025/Datasets_and_Benchmarks_Track — NeurIPS 2025 Datasets and Benchmarks Track poster_

### Official Review · Reviewer_EBRt · 2025-06-26

**Rating:** 5
**Confidence:** 5

**Summary:**

The paper introduces PartNeXt, a large-scale, high-quality 3D part dataset comprising about 24K textured meshes annotated with fine-grained, hierarchical part labels. The authors develop a fully web-based, dual-panel interface featuring connectivity-aware, bounding-box, and per-face selection tools. They also benchmark PartNeXt on two tasks—class-agnostic part segmentation and a new 3D part-centric question-answering (QA) task—demonstrating that state-of-the-art methods (e.g., PartField, SAMPart3D, SAMesh) struggle with leaf-level parts and that 3D-LLMs (PointLLM, ShapeLLM, 3DLLM) exhibit limited part-level reasoning. Finally, they show that training Point-SAM on PartNeXt yields substantial gains over training on PartNet, validating the dataset’s utility for advancing fine-grained 3D understanding.

**Dataset Code Accessibility:**

Yes

**Dataset Code Comments:**

The part annotations are uploaded to HuggingFace. Combined with the Objaverse dataset, it's readily usable for researchers to reproduce the results.

**Ethical Comments:**

The dataset is constructed from publicly available and licensed 3D datasets.

**Ethical Considerations:**

No, there are no or only very minor ethics concerns

**Final Justification:**

The rebuttal adequately addresses my concerns, and I believe the paper makes a solid contribution to the field of 3D generation.

**Limitations Weaknesses:**

- The dataset is primarily composed of common objects with relatively simple structures and well-defined parts, which may limit its applicability to more complex or ambiguous categories.
- While the annotation tool is a valuable contribution, making it open source would further benefit the community and support broader adoption.

**Strengths Contributions:**

- The paper presents a well-designed annotation tool that thoughtfully addresses practical challenges, significantly improving both annotation efficiency and accuracy.
- The dataset is manually labeled with a cross-review process, resulting in high-quality part annotations for approximately 24K samples—representing the first effort of this scale in the field.
- PartNeXt serves as a comprehensive benchmark for 3D part segmentation, enabling rigorous evaluation of existing and future methods.
- The authors validate the dataset through two distinct tasks, effectively demonstrating the limitations of current approaches and highlighting the utility of the new dataset.
- The paper provides thorough implementation details and offers insightful statistics to support reproducibility and further research.

---

> ### Author Rebuttal · Authors · 2025-07-30
>
> We sincerely appreciate your valuable and constructive feedback. We are pleased to see that you recognize the significance of PartNeXt to the community, including the PartNeXt dataset, the annotation platform, and the PartNeXt benchmark. Our work proposes a next-generation fine-grained and hierarchical 3D Part Dataset, along with a challenging part segmentation benchmark, and a part-centric question answering benchmark. We believe that Neurips Dataset and Benchmark Track is the optimal venue to publish our work and contribute to the community.
>
> Upon acceptance, **we will release**
> 1. The full PartNeXt dataset with detailed guidelines and library for loading, visualizing, and processing annotated data,
> 2. PartNeXt benchmark code with detailed guidelines, as well as the testset datalist used in our experiments and the results of the benchmark for community verification,
> 3. PartNeXt's annotation platform and codes.
>
> In the following, we will address your valuable and insightful feedback regarding PartNeXt’s limitations. **We would really appreciate it if you could raise the score given the resolved concerns.**
>
> # Clarification for Limitation 1: The Dataset only contains common objects with simple structure and will-defined parts.
>
> We acknowledge that PartNeXt focuses on common objects with relatively simple structures and well-defined parts. However, our primary target is to establish a high-quality 3D part-level dataset, the selection of clear structure objects with unambiguous part definitions ensures annotation precision, consistency, and efficiency. Therefore, PartNeXt leverages categories with clear structure and well-defined parts for annotation.
>
> However, as our annotation platform is designed for general object annotation, we can also annotate more complex object categories given corresponding predefined hierarchies. However, annotating open vocabulary objects with complex structures remains an open question, which we leave as future work.
>
> We hope PartNeXt will raise awareness in the community about the significance of high-quality fine-grained 3D part datasets, provide design experience in building annotation pipelines and tools, and inspire the community to explore the annotation of open-vocabulary part datasets.
>
> # Clarification for Limitation 2: Open source of the annotation platform
>
> Thank you for the recognition of our annotation system's value. We design the WebGL-based dual-panel annotation system to achieve part-level annotation with original texture preservation, effectively addressing internal component annotation challenges.
>
> Upon acceptance, we will release the source code of our annotation platform publicly.

---

> > ### Comment · Reviewer_EBRt · 2025-08-02
> >
> > Thank the authors for the rebuttal. It solves my concerns and I will keep the current score.

---

### Official Review · Reviewer_uXvi · 2025-06-28

**Rating:** 5
**Confidence:** 4

**Summary:**

The paper introduces PartNeXt, a large and important new dataset of over 23,000 textured 3D models with detailed, hierarchical part labels. It directly addresses the limitations of older datasets by including textures and using a more scalable annotation method. The authors prove its value by showing that it challenges even state-of-the-art models and significantly improves their performance when used for training.

**Dataset Code Accessibility:**

Yes

**Ethical Considerations:**

No, there are no or only very minor ethics concerns

**Final Justification:**

The rebuttal addresses my concerns, and i will keep this positive score.

**Limitations Weaknesses:**

1.The authors leverage GPT-4o to bootstrap the part hierarchies. While this is an innovative approach for scalability, it introduces a potential source of systematic, non-human-like bias. The paper states these hierarchies were "reviewed and refined by human experts," but it lacks detail on the extent of this refinement. It is unclear what percentage of the AI-generated hierarchies were accepted as-is versus those requiring significant modification.

2. The decision to select the 50 categories with the most objects introduces a significant popularity bias. As seen in Table 1, the dataset is heavily skewed, with categories like "Chair" (4277 models) and "Table" (2326 models) dominating, while others like "Buck" (8 models) or "WiFi" (8 models) are barely represented. This imbalance could lead to models trained on PartNeXt being over-fitted to common furniture items, limiting their generalizability to rarer object classes.

**Strengths Contributions:**

1.The dataset addresses a clear and important gap. The inclusion of high-quality textures and fine-grained, hierarchical part labels pushes the field beyond what was possible with previous datasets like PartNet.

2.The authors have clearly put significant thought into the annotation process. The fully web-based, dual-panel interface is a clever design that effectively addresses the challenge of annotating occluded and internal parts. This, combined with a flexible suite of face-selection tools, lowers the barrier for annotation and represents a notable contribution in its own right.

3.The experiment showing that Point-SAM trained on PartNeXt substantially outperforms its PartNet-trained counterpart is compelling evidence of the dataset's higher quality and diversity.

---

> ### Author Rebuttal · Authors · 2025-07-30
>
> We sincerely appreciate your valuable and constructive feedback. We are pleased to see that you recognize the significance of PartNeXt to the community, including the PartNeXt dataset, the annotation platform, and the PartNeXt benchmark. Our work proposes a next-generation fine-grained and hierarchical 3D Part Dataset, along with a challenging part segmentation benchmark, and a part-centric question answering benchmark. We believe that Neurips Dataset and Benchmark Track is the optimal venue to publish our work and contribute to the community.
>
> Upon acceptance, **we will release**
> 1. The full PartNeXt dataset with detailed guidelines and library for loading, visualizing, and processing annotated data,
> 2. PartNeXt benchmark code with detailed guidelines, as well as the testset datalist used in our experiments and the results of the benchmark for community verification,
> 3. PartNeXt's annotation platform and codes.
>
> In the following, we will address your valuable and insightful feedback regarding PartNeXt’s limitations. **We would really appreciate it if you could raise the score given the resolved concerns.**
>
> # Clarification for Limitation 1: Details of refinement on hierarchy
>
> In defining the hierarchies, all hierarchies were carefully refined manually. The refinement process involves:
> 1. Removing components that cannot exist in artist models (e.g., motherboards in keyboards),
> 2. Adding part variants,.
> 3. Adjusting hierarchies to better align with real-world objects and ease annotation.
>
> For a small number of categories with simple structures (e.g., keyboards, cups), minimal modifications were needed. However, for the majority of complex structures, detailed adjustments were made to meet our hierarchy design principles.
>
> Before the large-scale annotation starts, we select a sample dataset covering all categories for annotators to test the annotation system. During this testing phase, we collected feedback from the annotators to further optimize the hierarchy, aiming to cover as many object structures and parts as possible.
>
> # Clarification for Limitation 2: Imbalanced category data number
>
> The hierarchy for each category needs to be suitable for crowd-sourced annotation, allowing annotators to accurately understand and perform the annotation. While we can define well-defined hierarchies for certain specific categories, constructing hierarchies for general categories remains an open problem. We are actively working on this challenge as part of our future research.
>
> Therefore, in the initial phase, we selected 100 categories that are relatively easy to annotate and well-suited for hierarchical definition. After applying our CLIP-based data filtering strategy followed by manual refinement, the number of 3D models suitable for annotation becomes limited, and some categories contain only a small number of models. Therefore, the number of data in each category is imbalanced.
>
> To further expand PartNeXt, we are also attempting to collect additional data from the full ObjaverseXL dataset, aiming to cover a broader range of objects and categories, thus enhancing overall data diversity and addressing the imbalanced number of data in each category.

---

> > ### Comment · Reviewer_uXvi · 2025-08-04
> >
> > Thanks for the detailed response, i will keep the positive score.

---

### Official Review · Reviewer_X5Vo · 2025-06-29

**Rating:** 5
**Confidence:** 4

**Summary:**

This paper introduces PartNeXt, a large-scale 3D dataset consisting of over 23,000 textured 3D models annotated with fine-grained, hierarchical part labels across 50 object categories. The authors present novel tools for scalable and crowd-friendly annotation directly on textured meshes, leveraging AI (CLIP and GPT-4o) to assist in taxonomy definition and object selection. The dataset supports two benchmarks: class-agnostic part segmentation and 3D part-centric question answering (3D QA). Empirical results suggest existing methods struggle on these tasks, and training on PartNeXt improves segmentation performance.

**Additional Feedback:**

1.Explicit Limitation Discussion: Include a section discussing potential limitations, such as the uneven object distribution across categories, reliance on synthetic data, and generalizability to real-world scenarios. This would provide a balanced perspective and guide future research.
2.Real-World Applicability: Provide insights or experiments on how PartNeXt performs with real-world scanned data, addressing concerns about its synthetic focus. Case studies or comparisons could strengthen its practical relevance.
3.Category Distribution Analysis: Discuss the implications of uneven object distribution and consider strategies to mitigate the impact of low-sample categories, such as data augmentation or targeted data collection.

**Dataset Code Accessibility:**

Yes

**Dataset Code Comments:**

The dataset’s open access and could be downloaded from Huggingface.

**Ethical Considerations:**

No, there are no or only very minor ethics concerns

**Final Justification:**

The rebuttal resolves my concerns, and I appreciate the detailed response. I will maintain the positive score.

**Limitations Weaknesses:**

1.Lack of Limitation Discussion: The paper does not address its own limitations, as indicated in the NeurIPS checklist (Question 2: Answer [NA]). This omission overlooks potential issues like generalizability, data biases, or scalability for complex objects, which could mislead researchers about the dataset’s scope.
2.Synthetic Data Focus: PartNeXt primarily uses synthetic 3D models from sources like Objaverse, which may not fully represent real-world scanned data or uncontrolled environments. This could limit its applicability in practical scenarios.
3.Uneven Object Distribution: While PartNeXt includes 23,519 objects across 50 categories, the distribution is uneven, with some categories having very few objects (e.g., 8 for 'Cam', 9 for 'Door'). Compared to PartNet’s 26,671 objects across 24 categories (averaging ~1,111 objects per category), PartNeXt’s average of ~470 objects per category is lower, potentially limiting model robustness for underrepresented categories.

**Strengths Contributions:**

1.Enhanced Data Quality: Unlike PartNet’s untextured geometries, PartNeXt uses textured meshes, providing richer visual cues for fine-grained understanding. With 23,519 models and 350,187 annotated parts across 50 categories, it surpasses PartNet’s 24 categories, enabling broader applications.
2.Innovative Annotation System: The web-based interface, supported by AI tools like CLIP and GPT-4o, streamlines annotation, making it accessible to non-experts and enhancing efficiency. This approach supports large-scale data creation and potential community contributions.
3.Comprehensive Benchmarks: The paper introduces two  benchmarks: one for class-agnostic 3D part instance segmentation and another for 3D part-centric question answering. These tasks reveal deficiencies in existing models, particularly in fine-grained part understanding, and set a foundation for future 3D LLM research.
4.Clear Empirical Validation: Benchmarks convincingly show state-of-the-art models struggle with PartNeXt, demonstrating its utility.

---

> ### Author Rebuttal · Authors · 2025-07-30
>
> We sincerely appreciate your valuable and constructive feedback. We are pleased to see that you recognize the significance of PartNeXt to the community, including the PartNeXt dataset, the annotation platform, and the PartNeXt benchmark. Our work proposes a next-generation fine-grained and hierarchical 3D Part Dataset, along with a challenging part segmentation benchmark, and a part-centric question answering benchmark. We believe that Neurips Dataset and Benchmark Track is the optimal venue to publish our work and contribute to the community.
>
> Upon acceptance, **we will release**
> 1. The full PartNeXt dataset with detailed guidelines and library for loading, visualizing, and processing annotated data,
> 2. PartNeXt benchmark code with detailed guidelines, as well as the testset datalist used in our experiments and the results of the benchmark for community verification,
> 3. PartNeXt's annotation platform and codes.
>
> In the following, we will address your valuable and insightful feedback regarding PartNeXt’s limitations. **We would really appreciate it if you could raise the score given the resolved concerns.**
>
> # Clarification for Limitation 1: Lack of Limitation Discussion
>
> We appreciate the reminder to add the discussion of limitations. Currently, PartNeXt faces three limitations:
>
> 1. Limited Dataset Size. To ensure high-quality annotations, PartNeXt currently includes 23,519 models. We are actively working on expanding the dataset by incorporating more data from ObjaverseXL.
> 2. The need for predefinition of fine-grained hierarchy. Each category in PartNeXt requires a carefully designed fine-grained part hierarchy. This constraint limits our ability to annotate open-vocabulary datasets. We are exploring open-vocabulary part annotations through deeper integration with VLMs.
> 3. Plain part name annotations. Currently, PartNeXt provides only part names for each node. Introducing caption-level annotations for category and part could greatly enrich the semantic information of the PartNeXt dataset.
>
> We will add the detailed discussion on the limitations of PartNeXt in the revised paper.
>
> # Clarification for Limitation 2: Synthetic Data Focus
>
> Our annotation mainly focuses on synthetic data, as these data are typically artist-modeled, containing potential structure information (from connected components and scene graphs) and high-fidelity textures. Our annotation tools leverage these properties to enable efficient labeling.
>
> Our platform also supports annotation on real-world data with the help of hierarchy modification and face/box selection. However, real-world data usually comes from reconstruction and only contains surfaces or point clouds, lacking internal structure and connectivity information. This makes it difficult to fully leverage the selection tool of our annotation platform. Therefore, we do not showcase annotations on real data in this work. Nevertheless, it is important to emphasize that our platform still supports loading and annotating real-world data.
>
> # Clarification for Limitation 3: Uneven Object Distribution
>
> The hierarchy for each category needs to be suitable for crowd-sourced annotation, allowing annotators to accurately understand and perform the annotation. While we can define well-defined hierarchies for certain specific categories, constructing hierarchies for general categories remains an open problem. We are actively working on this challenge as part of our future research.
>
> Therefore, in the initial phase, we selected 100 categories that are relatively easy to annotate and well-suited for hierarchical definition. After applying our CLIP-based data filtering strategy followed by manual refinement, the number of 3D models suitable for annotation becomes limited, and some categories contain only a small number of models. Therefore, the distribution of data in each category is uneven.
>
> To further expand PartNeXt, we are also attempting to collect additional data from the full ObjaverseXL dataset, aiming to cover a broader range of objects and categories, thus enhancing overall data diversity and addressing the uneven object distribution.

---

> > ### Comment · Reviewer_X5Vo · 2025-08-05
> >
> > The rebuttal resolves my concerns, and I appreciate the detailed response. I will maintain the positive score.

---

### Official Review · Reviewer_gPhE · 2025-07-02

**Rating:** 5
**Confidence:** 4

**Summary:**

This work presents a next-gen 3D part segmentation dataset that extends the number of object categories from previous 24 to 50, improves fined-grained part hierarchy. The whole dataset has ~23K objects in 50 object categories, with ~350K parts annotated in total. The annotation interface makes human annotation much more intuitive and reliable, the annotation is done over the textured mesh faces, without any remeshing. The benchmark is comprehensive. Part segmentation and part-centric question answering tasks are used to benchmark stoa models, which shows the new challenges this new dataset brings. This dataset contributes an important testbed for 3d part understanding.

**Dataset Code Accessibility:**

Partly

**Dataset Code Comments:**

data is hosted on huggingface; benchmark code is not accessible yet

**Ethical Comments:**

in the released dataset, you should make dataset license explicitly available.

**Ethical Considerations:**

No, there are no or only very minor ethics concerns

**Final Justification:**

the rebuttal provides enough details to address what is unclear to me. I will maintain my rating.

**Limitations Weaknesses:**

1. why do you decide to curated similar number (~25K) of objects as PartNet when there are much more 3D datasets available nowadays? e.g. you started with 100 categories, but only selected 50 for annotation. 50 is better than 24, but still seems limited? Or in the part level, the additional 50 do not provide more diversity any more.
2. It seems you refined the part hierarchy with a set of design criteria. Is there any thoughts about backward compatible with PartNet, for example in the leaf-node level? e.g. for model comparison, or is it not aa issue.
3. "other" is used when it is not clear how to annotate a part, is there some statistics about "other"? are you using a iterative way to refine the part hierarchy after getting the feedback from "other" parts?
4. I could not find training details for the part segmentation experiments (sec4.1), e.g. data split, important hyperparams.
    - 4.1. why there is a *big* difference of mIoU btw SAMesh and PartField on some categories in table 2?
    - 4.2. is texture used as the model input (e.g. as point color)?
    - 4.3. do you plan to release the benchmark code for people to reproduce and as examples to start using PartNeXt?

minor
- L283, PointSAM[7] ref is wrong

**Strengths Contributions:**

- Compared with previous part datasets such as PartNet, PartNet-Mobility, PartObjaverseTiny, the presented PartNeXt is a new datasets that covers more object categories from not just shapenet but also ABO and 3D-FUTURE; refined part hierarchy with more carefully considered hierarchy definition (e.g. affordance-aware); and directly on original textured mesh instead of remeshed geometry.
- The introduced web-based annotation interface is intuitive to use, thus reducing the cost to annotate more objects with fine-grained parts. The attached video demonstrates the interface quite well. The authors make contributions in improved part hierarchy, dual-panel display, and three selection tools. The annotation is done directly on the original (textured) mesh without remeshing or cutting the mesh. The annotation tool can be used to further annotate more 3D mesh data in a more accessible way.
- Benchmark on this new dataset with two core part-related tasks: class-agnostic 3d part segmentation and part-centric question answering. The results demonstrate the new challenges that this dataset brings, stoa models still have large room to improve. Also trained one model (PointSAM) on different variations of part datasets to show the improved results when trained on PartNeXt over PartNet.
- Paper is very well-written.
- Lots of details are presented in the appendix.

---

> ### Author Rebuttal · Authors · 2025-07-30
>
> We sincerely appreciate your valuable and constructive feedback. We are pleased to see that you recognize the significance of PartNeXt to the community, including the PartNeXt dataset, the annotation platform, and the PartNeXt benchmark. Our work proposes a next-generation fine-grained and hierarchical 3D Part Dataset, along with a challenging part segmentation benchmark, and a part-centric question answering benchmark. We believe that Neurips Dataset and Benchmark Track is the optimal venue to publish our work and contribute to the community.
>
> Upon acceptance, **we will release**
> 1. The full PartNeXt dataset with detailed guidelines and library for loading, visualizing, and processing annotated data,
> 2. PartNeXt benchmark code with detailed guidelines, as well as the testset datalist used in our experiments and the results of the benchmark for community verification,
> 3. PartNeXt's annotation platform and codes.
>
> In the following, we will address your valuable and insightful feedback regarding PartNeXt’s limitations. **We would really appreciate it if you could raise the score given the resolved concerns.**
>
> # Clarification for Limitation 1: limited amount of data and categories
> The hierarchy for each category needs to be suitable for crowd-sourced annotation, allowing annotators to accurately understand and perform the annotation. While we can define well-defined hierarchies for certain specific categories, constructing hierarchies for general categories remains an open problem. We are actively working on this challenge as part of our future research.
>
> Therefore, in the initial phase, we selected 100 categories that are relatively easy to annotate and well-suited for hierarchical definition. After applying our CLIP-based data filtering strategy followed by manual refinement, the number of 3D models suitable for annotation becomes limited, and some categories contain only a small number of models. To ensure data quality and annotation efficiency, we finally select 50 categories for annotation, resulting in limited data number.
>
> To further expand PartNeXt, we are also attempting to collect additional data from the full ObjaverseXL dataset, aiming to cover a broader range of objects and categories, thus enhancing overall data diversity.
>
> # Clarification for Limitation 2: Backward compatible with PartNet
> For the categories present in PartNet, our hierarchy builds upon the original structure by adding new nodes and making structural adjustments. Since both PartNeXt and PartNet provide template hierarchies, users can establish the mapping between the overlapping nodes of the two datasets according to their use of the dataset.
>
> # Clarification for Limitation 3: Statistics on the  number of "Other" nodes and iterative refinement on hierarchy
>
> We provide statistics on "Other" here.
>
> | Number of Other | 0     | 1     | 2    | 3    | 4    | 5    | >5   |
> |-----------------|-------|-------|------|------|------|------|------|
> | Percentage %    | 82.22 | 13.82 | 1.77 | 0.30 | 0.38 | 0.20 | 1.30 |
>
> Before the large-scale annotation starts, we select a sample dataset covering all categories for annotators to test the annotation system. During this testing phase, we collected feedback from the annotators to further optimize the hierarchy, aiming to cover as many object structures and parts as possible.
>
> However, for rare parts, it was difficult to receive timely feedback. Therefore, upon completion of all data annotation, we initiated a revision round to address "Other" instances within the annotation structure. We filtered all annotated data containing more than 2 "Other" nodes and requested annotators to rename these "Other" nodes and place them at the appropriate positions within the hierarchical structure. During these modifications, our annotation system enabled annotators to efficiently rename and drag "Other" nodes either in batches or individually, making these changes convenient to complete.
>
> The final statistics demonstrate that the number of "Other" instances is low after revision.
>
> # Clarification for Limitation 4: Experiment details in Segmentation Benchmark
>
> In our segmentation experiments, to ensure fairness, we utilized publicly available code for the comparative methods (SAmesh, PartField, SAMPart3D), employing their publicly released checkpoints and using the original hyperparameters specified in their papers.
>
> Regarding dataset splits: since all three compared methods are zero-shot, we select 200 models from our dataset as the test set for evaluation.
>
> We will open-source all the source code for our benchmark, and also publish the list of the 200 models used in the segmentation benchmark.
>
> ## Clarification for Limitation 4.1: Big difference of mIoU between SAMesh and PartField on some categories
>
> This corresponds to an interesting fact we observed in experiments: SAMesh and PartField exhibit significant performance differences in certain categories. After examining the segmentation results, we found that PartField excels at segmenting parts with accurate semantic boundaries, but struggles to capture fine-grained components. In contrast, SAMesh leverages the geometric connectivity in objects to successfully segment these small structures. However, it shows limited capability with parts that aren't separately modeled in the original mesh. Therefore, SAMesh achieves exceptionally high performance on categories with small components (e.g., beds with feet or frame rods), while its accuracy decreases notably for categories lacking distinct part modeling (e.g., bottles where the neck and body come from a whole structure). We will provide the segmentation results from our experiments for community verification.
>
> ## Clarification for Limitation 4.2: Whether Texture as model input
>
> Regarding texture input, we follow each segmentation model's native settings. SAMesh utilizes textureless meshes, PartField uses colorless point clouds, sampled from mesh surfaces, SAMPart3D requires point clouds with color and normal attributes, so we sample colorized point clouds from textured meshes as input. We will add this detail to the revised paper.
>
> ## Clarification for Limitation 4.3: Release of the benchmark code
>
> As clarified above, upon acceptance, we will release the complete benchmark code for the community to verify our benchmark results, and provide detailed instructions for using the PartNeXt dataset.
>
> # Clarification for Minor issues
>
> We will fix the wrong reference, and add the dataset license.

---

> > ### Comment · Reviewer_gPhE · 2025-08-06
> >
> > Thanks for your detailed response. I will keep my rating.

---

### Decision · Program_Chairs · 2025-09-18

**Decision:**

Accept (poster)

**Comment:**

The paper proposes a 3D part dataset (PartNeXt) consisting of 23.5K 3D meshes from Objaverse, ABO, and 3DFuture.  Each 3D mesh is hierarchically decomposed into parts (with GTP-4o used to suggest the part hierarchy) and manually annotated.  The main contributions of the work are 1) the annotation framework, 2) the dataset, and 3) a set of experiments evaluating recent part segmentation work on the class-agnostic 3D part instance segmentation and part-centric 3D question answering.

All four reviewers were positive on the work, finding the annotation framework and dataset to be a valuable contribution with experiments demonstrating the usefulness of the dataset.  Reviewers also noted that the paper to be well-written [gPhE] and annotation/dataset details are provided in the appendix [gPhE].

One of the main weaknesses pointed out by reviewers is the limited size of the dataset (number of assets is comparable to PartNet, and only 50 categories is covered) [gPhE].  Reviewers also had questions regarding some design choices [gPhE,uXvi,EBRt] and experimental details [gPhE], and lack of a discussion of limitations [X5Vo].  Despite these concerns, reviewers are positive on the work.

The AC agrees that the work is a valuable addition to the landscape of 3D datasets, and appreciate the effort the authors put into curating this dataset, and recommend the work for acceptance.  The authors should incorporate clarifications and response to the reviewer into the main paper, in particular the inclusion of a discussion of the limitations of the work.  The AC also recommend the authors check the accuracy of some of the claims (e.g. "annotations in PartNet are performed on untextured, remeshed geometries" - as PartNet is the basis for some models for PartNetMobility where some of the texture information is retained, implying that some models are annotated with textured geometries).